# Long-duration electricity storage needs for coping with Dunkelflaute events in Europe

Martin Kittel [1,2] ✉, Alexander Roth [1,3] & Wolf-Peter Schill [1]

Coping with prolonged periods of low availability of wind and solar power, also referred to as variable renewable energy droughts or "Dunkelflaute", emerges as a key challenge for realizing decarbonized energy systems based on renewable energy. Here we investigate the role of long-duration electricity storage and geographical balancing through transmission in dealing with such events in Europe, combining a time series analysis of renewable availability with power sector modeling of 35 historical weather years. We find that extreme droughts define long-duration storage operation and investment. Assuming policy-relevant interconnection, the least-cost system in our model capable of coping with the most extreme event requires 351 terawatt hours long-duration storage capacity, corresponding to 7% of yearly European electricity demand. While nuclear power can partially reduce storage needs, the storage-mitigating effect of fossil backup plants in combination with carbon removal is limited. Policymakers and system planners should prepare for a rapid expansion of long-duration storage to safeguard the renewable energy transition in Europe.

To mitigate climate change and to meet international commitments, the European Union aims to achieve net zero emissions of greenhouse gases by 2050. The power sector will play a central role in a decarbonized economy. Massively expanding renewable energy sources would allow for a rapid substitution of fossil fuels in the power sector as well as in the industry, transport, and heating sector via electrification, a widely recognized decarbonization strategy also referred to as "sector coupling"[1–5]. The potentials of firm renewable energy sources, such as hydro- or bioenergy, are limited in most European countries. In contrast, wind and solar power have vast expansion potentials and promise declining costs[6,7]. Therefore, these variable renewable energy (VRE) technologies are likely to form the backbone of transitioning to net zero in most European countries[1,8]. With a rising share of VRE, the European power sector becomes increasingly exposed to weather variability[9,10]. This has spurred a debate in the energy policy domain about the security of supply[11–20]. Of particular concern are long-lasting extreme weather events, also referred to as "VRE droughts" or "Dunkelflaute", which

are characterized by an overall very low availability of VRE sources[21–23].

Previous research has shown that for growing shares of VRE increasing flexibility would not only enable a least-cost energy system, but also required to deal with imbalances in the power sector[24–29]. This can be provided by different types of electricity storage, demand response, or cross-border transmission. Spatial flexibility allows balancing of regional variations in demand and variable renewable supply by interconnecting countries via electricity and hydrogen grids[30,31]. In Europe, such cross-border interconnection can mitigate energy storage needs, particularly by balancing differences in wind power availability across countries[32]. Yet, it is unclear to what extent this storage-mitigating effect of geographical balancing persists during extreme pan-European VRE droughts of synoptic scale affecting many countries simultaneously[33]. Besides interconnection, a wide range of flexibility technologies can help to cope with VRE droughts. Particularly long-duration storage, sometimes also referred to as "ultra-long-duration storage"[34], is a central option as it can shift renewable surplus energy to

[1]German Institute for Economic Research (DIW Berlin), Department of Energy, Transportation, Environment, Anton-Wilhelm-Amo-Strasse 58, 10117 Berlin, Germany. [2]Technical University Berlin, Digital Transformation in Energy Systems, Einsteinufer 25 (TA 8), 10587 Berlin, Germany. [3]Bruegel, Rue de la Charité 33, 1210 Saint-Josse-ten-Noode, Belgium. ✉e-mail: mkittel@diw.de

periods of low wind and solar availability over long time scales, i.e., several days up to seasons[34–36]. Low energy capital costs are key to make long-duration storage viable in energy systems that heavily rely on wind and solar power[37–40]. From a techno-economic perspective, the production of hydrogen or its derivatives using renewable electricity, followed by storage and reconversion to electricity, currently appears most promising for enabling long-duration energy storage[36,41–43].

In this paper, we analyze how VRE droughts impact the operation and sizing of long-duration storage in a fully renewable European power sector. We further investigate how varying levels of geographical balancing via interconnection affect long-duration storage investments in the least-cost system. We also examine how different technologies interact during extreme VRE droughts and assess the role of nuclear power and fossil fuel-based backup capacity with emission abatement.

While there is growing research interest in VRE droughts and their power sector implications, respective analyses vary in subject as well as temporal and spatial scope[22]. One strand of the literature focuses on VRE drought characterization based on time series of wind speeds, solar irradiation, or normalized renewable availability factors, also referred to as capacity factors. This includes analyses of historical or future wind droughts for Germany[44], the North Sea[45], Ireland[46], or the UK[47–50]. These studies typically focus on frequency-duration distributions, return periods, or spatio-temporal correlations. Alternatively, wind droughts have been studied as VRE anomalies, i.e., cumulative deviations from climatological means or other reference profiles, on a global scale[51] or for the Netherlands[52]. Combining wind and solar power in renewable technology portfolios can mitigate drought characteristics within regions. This portfolio effect has been studied for Europe[21,33,53–56], the US[57], or individual countries, such as Germany[58,59], Hungary[60], India[61], and Japan[62]. In addition, the complementary of wind and solar power across regions further reduces extreme drought severity. This balancing effect has been illustrated for Europe[33,55,56,58] and the US[57]. Another literature strand explores positive residual load events, which relate to periods where VRE generation falls short of electric demand[22]. These periods indicate the need for system flexibility in general or, specifically, for a technology that supplies energy during such events. Positive residual load events have been studied for Europe[21,63–66], a subset of European countries (Norway, France, Italy, Spain, and Sweden)[23], Germany[67,68], Northern Italy[69], Africa[70], and the US[71]. Further, VRE droughts that cause power sector stress events can be detected in energy system models through shadow prices reflecting the value of stored energy or transmission grid capacity, extreme electricity prices, resource adequacy metrics, or emission patterns, as shown for Europe[72] or the US[73–76].

Energy system models increasingly include multiple sectors and energy carriers to explore future scenarios of sector-coupled energy systems[77,78]. Due to increasing complexity and computational burden, these models are often solved for only a single or a limited number of weather years. However, optimal energy model outcomes vary substantially across years in terms of optimal dispatch and investment in generation and storage capacities, transmission grid capacities, electricity demand, prices, levelized costs of electricity, or emissions. This is well-documented for single countries, such as the UK[9,79–82] or Ireland[82], for overall Europe[10,83–87], or for the US[36,88,89]. For Europe, the total system costs of a future net-zero energy system vary between ± 10% depending on the year used for system design[87]. A general finding of this literature strand is that energy systems modeling based on only one or a few weather years identifies system configurations that may lead to suboptimal capacity choices or operational inadequacies. Extreme renewable drought events, which vary substantially across years and regions, are likely to contribute significantly to these inter-annual variations in energy system modeling[33].

While there is growing interest in the meteorology and energy systems analysis domains on the impact of weather variability on renewable energy systems[76,90], the literature lacks a distinct analysis of the impact of extreme renewable droughts on energy storage needs, and how spatial system flexibility may alleviate these.

Here we combine two open-source methods to investigate how VRE droughts impact optimal long-duration storage investments in a least-cost configuration of a fully decarbonized European power sector. We use renewable time series analysis for VRE drought identification and a power sector model for determining least-cost operation and deployment decisions across a summer-to-summer planning horizon. We further quantify how much long-duration storage could be avoided with different degrees of electricity and hydrogen exchange between countries. We determine the long-duration storage need for dealing with extreme renewable droughts considering policy-oriented European interconnection levels, what "no-regret" long-duration storage capacity remains for a scenario with unconstrained geographical balancing of such events, and how much these results vary across years in our model setup. In doing so, we also shed light on appropriate weather year selection for modeling weather-resilient energy system scenarios in Europe. We additionally illustrate how different types of flexibility options interact for coping with extreme renewable droughts and explore what role nuclear or fossil power plants with emission abatement could play in mitigating storage needs.

## Results

### Extreme renewable energy droughts coincide with major discharging periods of long-duration storage

One of the principal insights from our time series and model-based analysis is that the operation of long-duration storage coincides with VRE droughts. Figure 1 illustrates drought patterns of simulated renewable technology portfolios with policy-relevant capacity mixes from the Ten Year Network Development Plan (TYNDP) 2022, which are retrieved from the renewable time series analysis. The figure further shows the least-cost long-duration storage operation for the weather year 1996/97, in which we find the most extreme pan-European drought in the data (Supplementary Fig. 1). Notably, several European countries are affected simultaneously[33]. The highlighted extreme drought events consist of sequences of shorter but more severe droughts within contiguous periods of well-below-average renewable availability. Such events may last up to several months and span across the turn of years.

During these long-lasting events, average renewable availability is well below its long-run mean, but still remains well above zero. This means that long-duration storage is not necessarily required to continuously discharge during the entire drought event. However, these longer events comprise shorter periods with significantly lower availability, lasting multiple days or even two to three weeks. During these periods, storage is often discharged at full capacity and the storage state-of-charge declines strongly, particularly when co-occurring with peak-demand periods.

Extreme events typically occur in winter, leading to significant long-duration discharge periods. Countries that heavily rely on wind power, such as the UK, may face the most extreme renewable technology portfolio events also in summer, driven by wind droughts that are generally more pronounced in summer in Europe[33,47]. Yet, electricity demand in the UK is much higher in winter than in summer, similar to other central and northern European countries. This seasonal demand effect by far outweighs the differences between summer and winter droughts. Consequently, UK's major storage discharging period coincides with the winter drought, while the slightly larger summer drought hardly affects the storage state-of-charge.

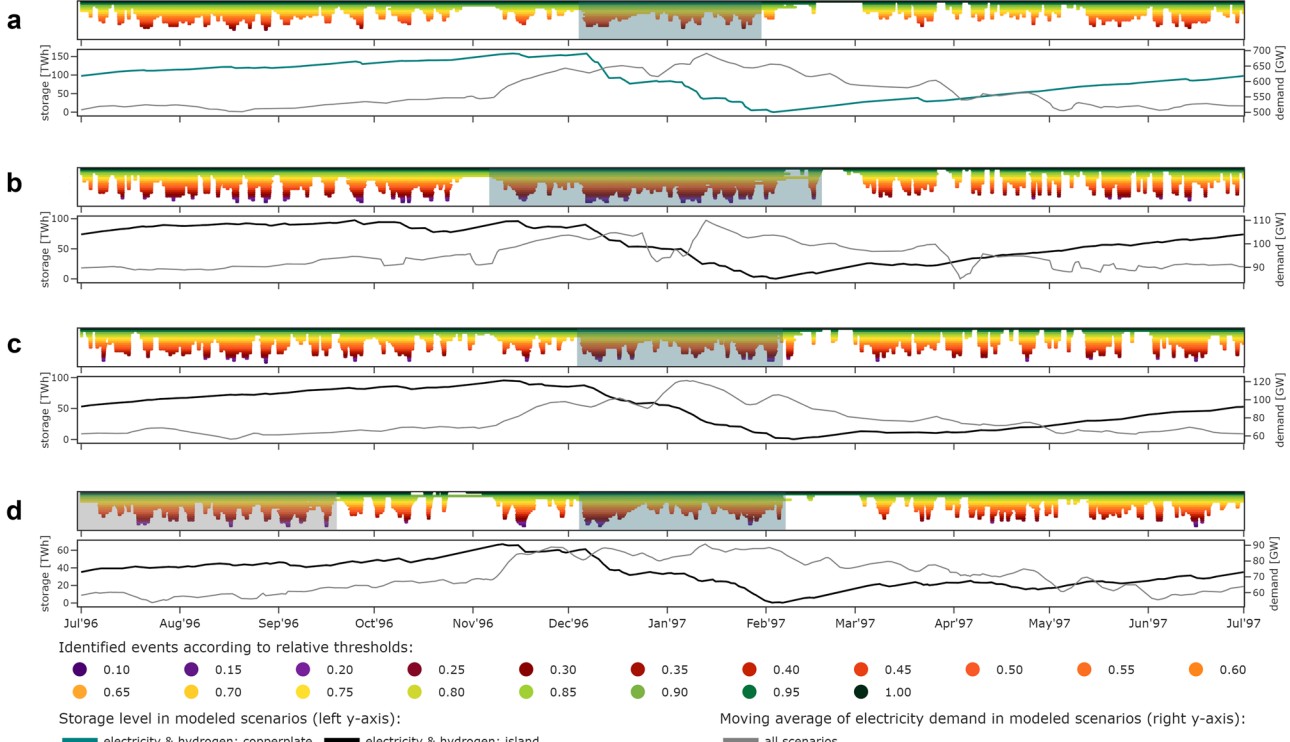

**Fig. 1 | Simulated drought events, electricity demand, and least-cost state-of-charge of long-duration storage in winter 1996/97.** The top row of each panel shows the identified drought patterns lasting longer than 12 h across all color-coded thresholds, with the most extreme drought events occurring in winter (teal boxes) or throughout the year (gray box in panel **d**). The bottom row of each panel displays the associated exogenous smoothed demand profiles used in the optimization and the resulting least-cost storage state-of-charge levels. Panel **a** corresponds to the pan-European copperplate scenario with unconstrained energy exchange across Europe, **b** to Germany as an island system, **c** to France as an island system, and **d** to the United Kingdom as an island system.

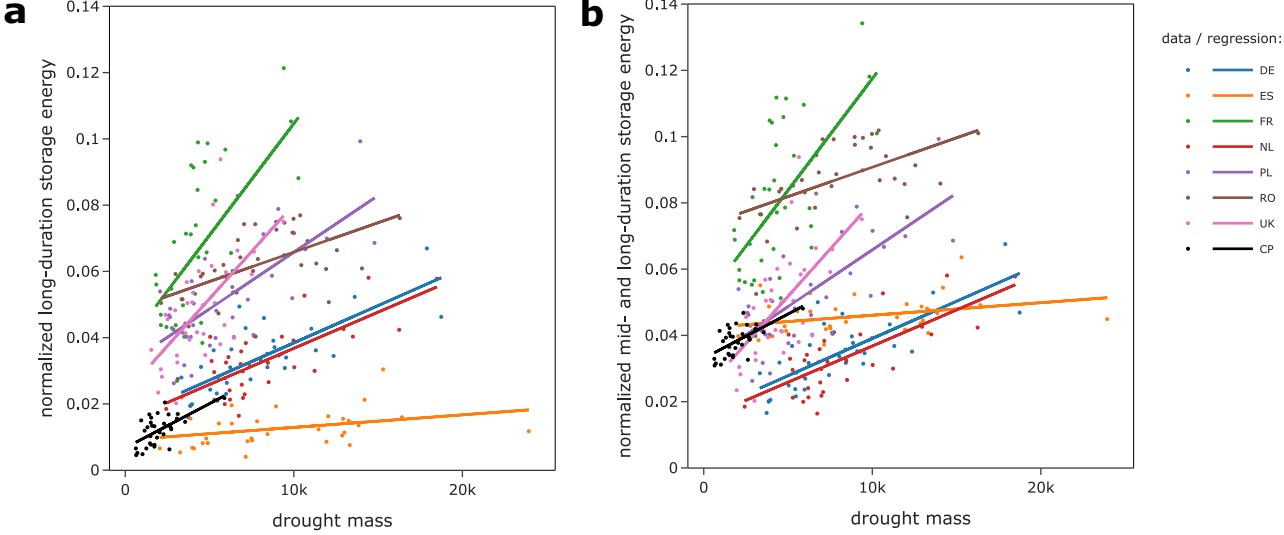

**Fig. 2 | Correlation of the severity of most extreme winter drought events (drought mass) and normalized storage energy capacity.** For comparison, we normalize the least-cost storage energy with the annual demand for electricity (including electrified heating) and hydrogen. For illustration, we exclude countries with least-cost storage energy below 5 TWh and countries with binding storage expansion potential constraints. Supplementary Fig. 3 shows the unfiltered regression results. We further include the pan-European copperplate scenario (CP). **a** Long-duration storage only. **b** Mid- and long-duration storage.

## Droughts drive long-duration storage energy capacity

For each modeled weather year, there is also a clear positive correlation between the most extreme renewable droughts in a given country and that country's long-duration storage energy need (Fig. 2a). We find that this applies to nearly all European countries when modeled as energy islands and also to the pan-European copperplate scenario.

The correlation between the most extreme winter drought events and least-cost storage capacities differs between countries. In Central European countries, such as France or the UK, storage capacity increases substantially in years with more severe droughts, illustrated by the steep slope of the fitted regression lines in Fig. 2. This effect is because electricity demand has a pronounced winter peak in these countries (Fig. 3).

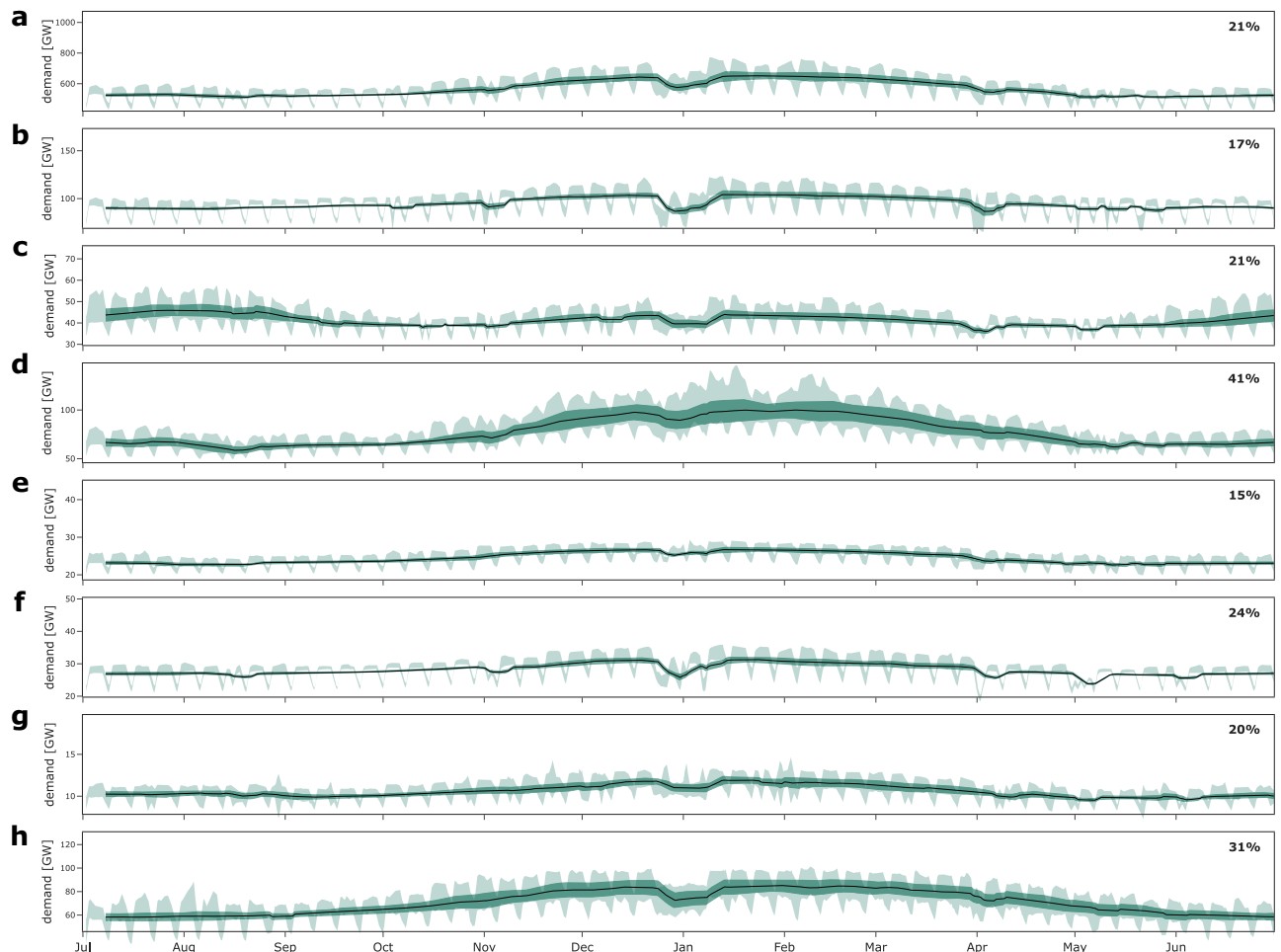

**Fig. 3 | Demand seasonality across regions.** The figure shows climatological mean demand as a bold line over all weather years using a moving average over a window of 168 hours (resulting in the blank first week) as a line, the standard deviation range as an area (*mean ± stddev*, dark green), the difference between the climatological minimum and maximum as an area using a moving average over a window of 24 hours (light green), and regional difference between the minimum and maximum climatological mean normalized by the maximum in the upper right corner (denoted as percentage). For comparison, each vertical axis is scaled to show its range if demand seasonality in this region was as pronounced as in France (panel **d**). Demand seasonality is particularly pronounced in France both in terms of level but also variance during winter due to high shares of electrified heating. Panel **a** corresponds to the pan-European copperplate scenario, **b** to Germany, **c** to Spain, **d** to France, **e** to the Netherlands, **f** to Poland, **g** to Romania, and **h** to the United Kingdom.

In contrast, more severe droughts hardly increase storage capacities in other countries, particularly in Romania or Spain, driven by a less seasonal or even summer-peaking electricity demand in such countries. Supplementary Fig. 4 shows how the storage-defining effect of extreme droughts slightly diminishes when removing the effect of demand seasonality in a stylized setting of the Germany power sector. Normalized long-duration storage energy needs are generally much lower in Spain than those in Central or Northern European countries. One driving factor for this are significant hydro reservoir and pumped hydro storage capacities in Spain, which substitute long-duration storage capacities for dealing with extreme droughts to some extent. A similar finding applies to the pan-European copperplate scenario. These effects lead to a lower sensitivity of long-duration storage needs to increasingly severe droughts, visible as lower regression line intercepts and, in the case of Spain, a relatively moderate slope of the regression line. Considering both hydro and long-duration storage capacities for the regression reduces these effects (Fig. 2b).

In the pan-European copperplate scenario, the lower drought mass show that droughts are notably less severe than in individual countries, resulting in significantly lower normalized storage needs. This is due to a geographical balancing effect, which spatially smooths renewable generation and demand patterns across all of Europe,

thereby mitigating extreme droughts[32,33], and reservoir and pumped hydro storage balancing renewable droughts without transmission constraints across Europe, particularly the large-scale capacities in Scandinavia.

In Section "Additional insights on the correlation between renewable droughts and long-duration storage energy" in the Supplementary Information, we discuss why the correlation between the most extreme winter drought event and the least-cost long-duration storage size for the same year is not perfect. We also provide results based on alternative capacity mixes with lower shares of solar photovoltaics (PV) for drought mass computation.

## Least-cost long-duration storage capacities decrease with geographical balancing

Interconnection between countries via electricity and hydrogen grids provides spatial flexibility to the power sector for coping with VRE droughts. To systematically assess this relationship, we analyze four scenarios with varying degrees of interconnection. Fig. 4 shows that increasing degrees of interconnection reduce aggregate storage energy needs across all scenarios.

In scenario (1), we model a counterfactual in which every country operates as an energy island, excluding the possibility of

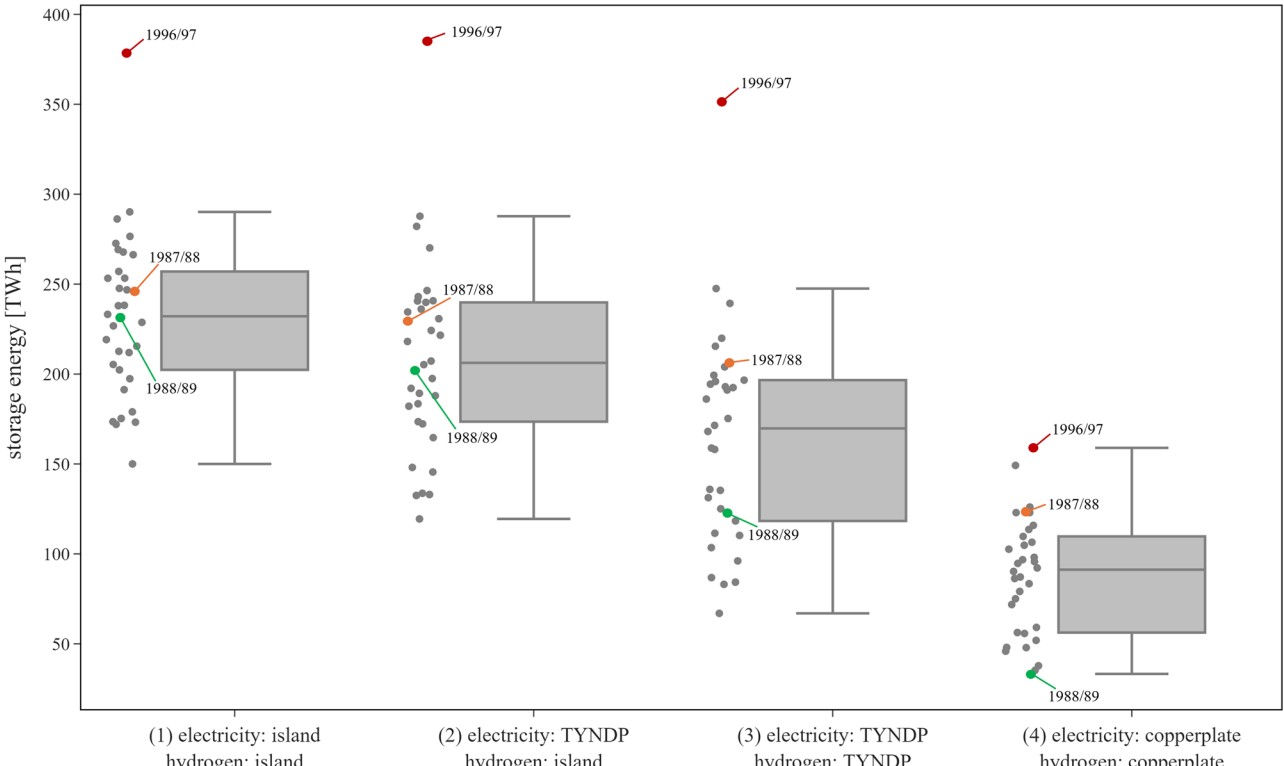

**Fig. 4 | Least-cost long-duration storage energy capacity aggregated across all countries for all modeled weather years and interconnection scenarios.** Each dot refers to one weather year modeled independently of other weather years. The center line denotes the median, box limits indicate the interquartile range (Q1-Q3), whiskers extend to 1.5 × the interquartile range, and points beyond the whiskers represent outliers. The year with the highest long-duration storage need is 1996/97 (red). The year that benefits most from rising interconnection capacity in terms of decreasing long-duration storage investments is 1988/89 (green). The year that benefits the least from increasing interconnection is 1987/88 (orange). Supplementary Fig. 8 illustrates the impact of interconnection on the ranking of weather years in terms of least-cost long-duration storage energy.

exchanging electricity or hydrogen with its neighbors. The resulting median (maximum) long-duration storage energy capacity aggregated across all modeled countries is 232 (378) terawatt hours (TWh). This corresponds to around 4.7 (7.6)% of the yearly European electricity demand. Scenario (2) allows for policy-oriented TYNDP 2022 cross-border exchange of electricity, which reduces (increases) the median (maximum) long-duration storage energy capacity to 206 (385) TWh, or 4.2 (7.8)% of yearly demand. The slight increase in maximum storage investments is mainly driven by an increase in offshore wind generation in Belgium, which is transmitted to and stored in additional long-duration storage capacities in the Netherlands, while in Belgium the long-duration storage energy capacity remains at its expansion limit. Additionally including hydrogen exchange within policy-oriented limits in scenario (3) further decreases the median (maximum) long-duration storage energy to 170 (351) TWh, equivalent to 3.4 (7.1)% of yearly demand. Scenario (4) models the counterfactual of unconstrained exchange of electricity and hydrogen between countries. This results in the lowest storage energy investments with a median (maximum) value of 91 (159) TWh, corresponding to 1.8 (3.2)% of yearly demand. Increasing interconnection capacity enables more pronounced geographical balancing, which smooths the impact of storage-defining renewable droughts, resulting in decreasing least-cost storage capacity[33]. Higher levels of interconnection capacities reduce the need for long-duration storage energy capacity both on an aggregate level and in most countries (Supplementary Fig. 12). If the expansion of the electricity or hydrogen networks in scenarios (2) or (3), as envisaged by European transmission system operators, cannot be realized, our results suggest that least-cost long-duration storage energy capacities will be between those of scenarios (1) and (2).

In our model, least-cost storage deployment shows a substantial degree of inter-annual variation. This is driven by differences in renewable energy droughts and demand patterns across years (compare Fig. 3). Compared to the island setting in scenario (1), the inter-annual variation is higher in scenarios (2) and (3) with policy-oriented geographical balancing. This is because the correlation of storage-defining droughts in individual countries varies across space and weather years[33]. When drought events do not occur simultaneously across countries, more pronounced geographical balancing can be leveraged. This reduces long-duration storage energy investments, which decreases the lower bound of the range. This effect is even more pronounced when including cross-border exchange hydrogen in scenario (3). As all drought events are balanced to the fullest extent possible in the copperplate scenario (4), we find the lowest inter-annual variation in this case.

Among the weather years in our data, 1996/97 marks the period with the highest aggregated long-duration storage energy investments across all interconnection scenarios in the least-cost system configurations. This is in line with our renewable availability time series analysis, which finds that the most extreme pan-European drought affects many countries simultaneously (Supplementary Fig. 1). In our model, the reducing effect of geographical balancing on long-duration storage investments remains limited for policy-oriented interconnection levels in scenarios (2) and (3) in 1996/97. This is because the storage-defining drought event in this year is temporally highly correlated across many countries and also coincides with demand peaks in several countries (see teal boxes and gray lines in Fig. 5). In contrast, the unconstrained balancing in the copperplate scenario (4) substantially mitigates overall storage investments, as energy can be imported from countries that are not

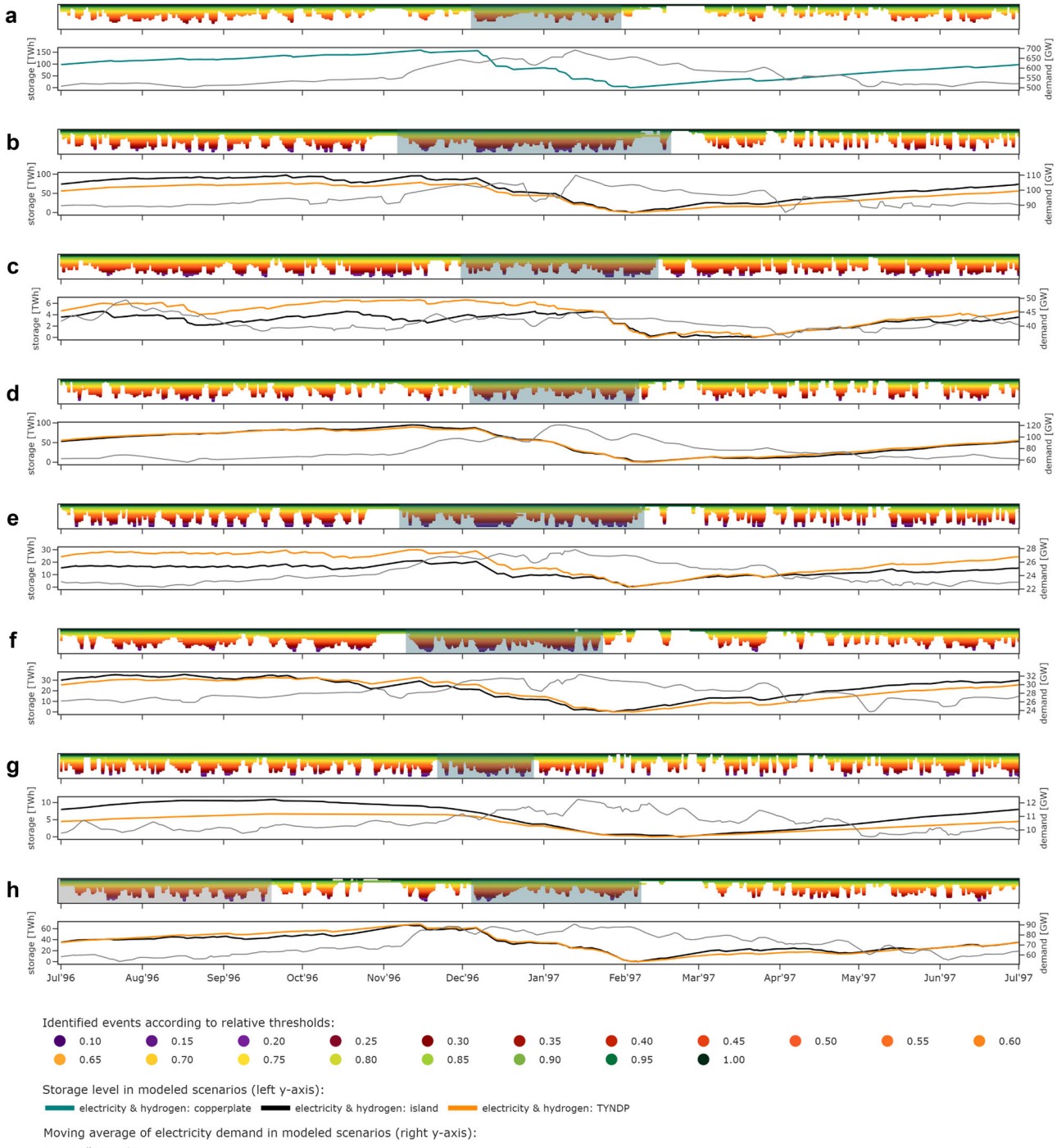

**Fig. 5 | Simulated drought events, electricity demand, and least-cost state-of-charge of long-duration storage in winter 1996/97 in countries with highest long-duration storage energy capacities.** The top row of each panel shows the identified drought patterns lasting longer than 12 hours across all color-coded thresholds, with the most extreme drought events occurring in winter (teal boxes) or throughout the year (gray box in panel **h**). The bottom row of each panel displays the associated exogenous smoothed demand profiles used in the optimization and the resulting least-cost storage state-of-charge levels for isolated countries modeled within the interconnection scenario (1), for policy-oriented interconnection levels in scenario (3), or the pan-European copperplate in scenario (4). Panel **a** corresponds to the pan-European copperplate scenario, **b** to Germany, **c** to Spain, **d** to France, **e** to the Netherlands, **f** to Poland, **g** to Romania, and **h** to the United Kingdom.

or less affected by renewable droughts. In other years, the effects of interconnection may play out differently. For instance, geographical balancing has a particularly strong storage-mitigating effect in 1988/89, as the largest droughts in major countries hardly overlap and co-occur with periods of high demand only to a limited extent (Supplementary Fig. 6). In contrast, the storage needs in 1987/88 seem less affected by interconnection. Its relative position in the ranking of years even slightly increases with increasing interconnection. This can be explained by a large temporal overlap of droughts in individual countries (Supplementary Fig. 7). Supplementary Fig. 8 additionally illustrates how higher interconnection levels increasingly affect the ranking of weather years.

In the least-cost system configuration, the maximum long-duration storage energy capacity of 159 TWh in scenario (4) occurs in 1996/97,

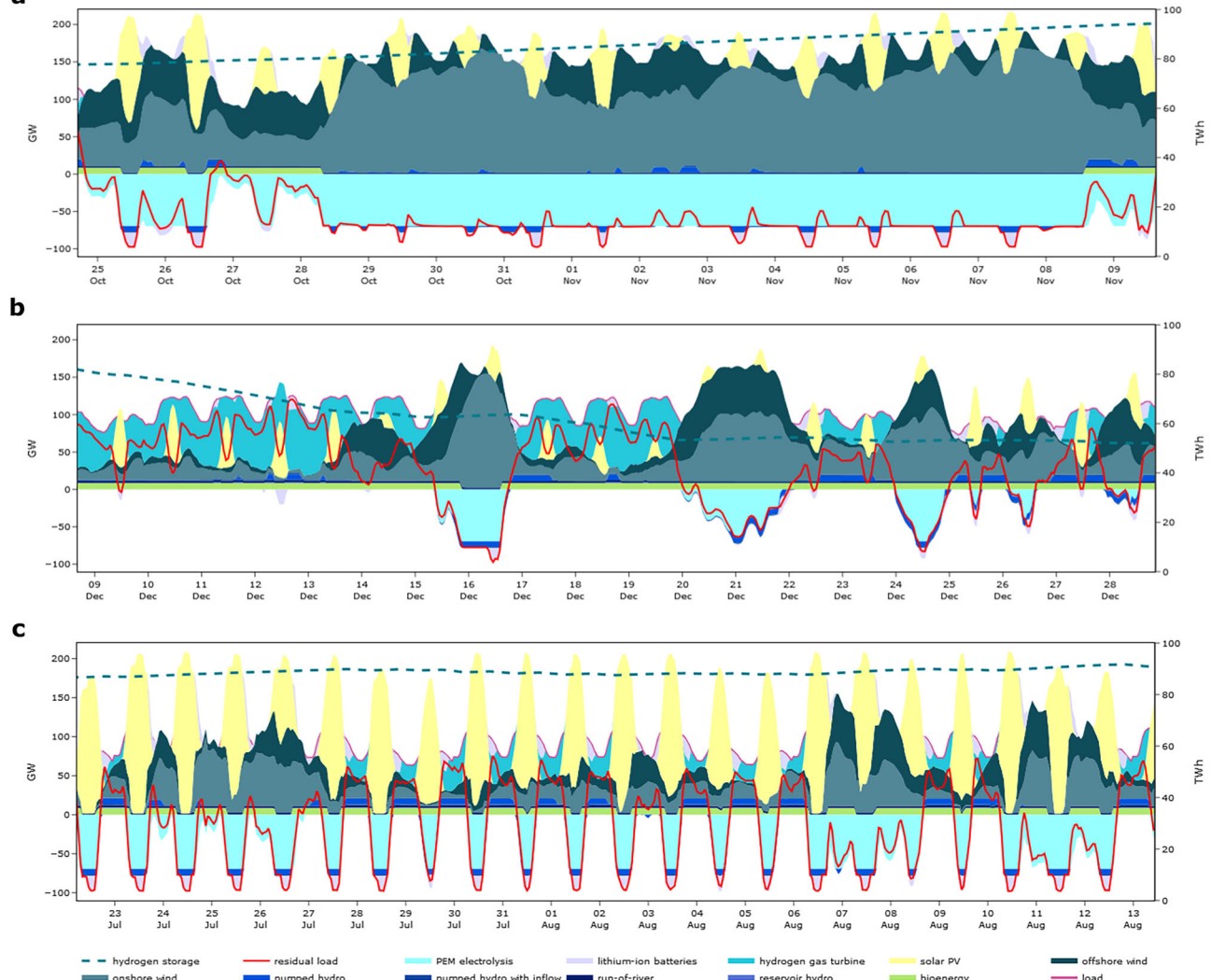

**Fig. 6 | Least-cost power sector operation in Germany for the weather year 1996/97.** The positive part of the left y-axis relates to generation and storage discharge, and its negative part to electricity demand and storage charge. The right y-axis refers to the long-duration storage state-of-charge. For illustration, we focus on scenario (1) excluding cross-border exchange of electricity or hydrogen. **a** Long-duration charging period before an extreme drought. **b** Long-duration discharge period within an extreme drought. **c** Long-duration storage follows diurnal solar PV pattern during summer.

which comprises the most extreme VRE drought in the data. This storage capacity is required to balance the remaining energy deficit of this event under the assumption of unlimited cross-country exchange of electricity and hydrogen. That is, storage capacity needed to deal with this event cannot be compensated any further by additional interconnection, given that geographical balancing has already been leveraged across all of Europe to the greatest extent possible. Hence, in our model, 159 TWh or 3.2% of the yearly European electricity demand, is the minimum need for long-duration storage capacity for a reliable, fully renewable European energy system. However, since unconstrained geographical balancing cannot be realized in reality, scenario (3) shows the most policy-relevant long-duration storage need that is required for dealing with the most extreme simulated Dunkelflaute event in the data. In this scenario, long-duration storage capacities are much higher, amounting to 351 TWh or 7.1% of the yearly demand.

### Interactions of different flexibility options for coping with extreme droughts

In our model, we find complex interactions between long-duration storage and other flexibility options in the power sector. The least-cost

dispatch patterns in exemplary weeks in Germany and Spain for the weather year 1996/97 in Fig. 6 illustrate such interactions.

Short-duration flexibility options, such as batteries and pumped hydro storage, generally have low costs for power capacity but high costs for energy capacity. This means that supplying power is relatively inexpensive, but storing energy is costly. In contrast, long-duration flexibility options, such as hydrogen-based electricity storage, exhibit opposite cost structures: electrolysis and hydrogen turbines, used for long-duration storage charging and discharging, incur high power capacity costs, while long-duration energy capacity costs are low. Hydro reservoirs and bioenergy have similar cost structures and serve comparable roles as long-duration storage. Given these differences in cost structures, the dispatch patterns of these flexibility options show different effects in our model-based scenarios that enable coping with extreme VRE droughts at the lowest possible cost.

Our model finds that during charging periods of the long-duration through electrolysis, batteries may charge and discharge, or bioenergy and hydro reservoirs generate electricity to maximize electrolyzer utilization, even if residual load is negative (Fig. 6a). In doing so, these short-duration flexibility options enable a continuous operation of

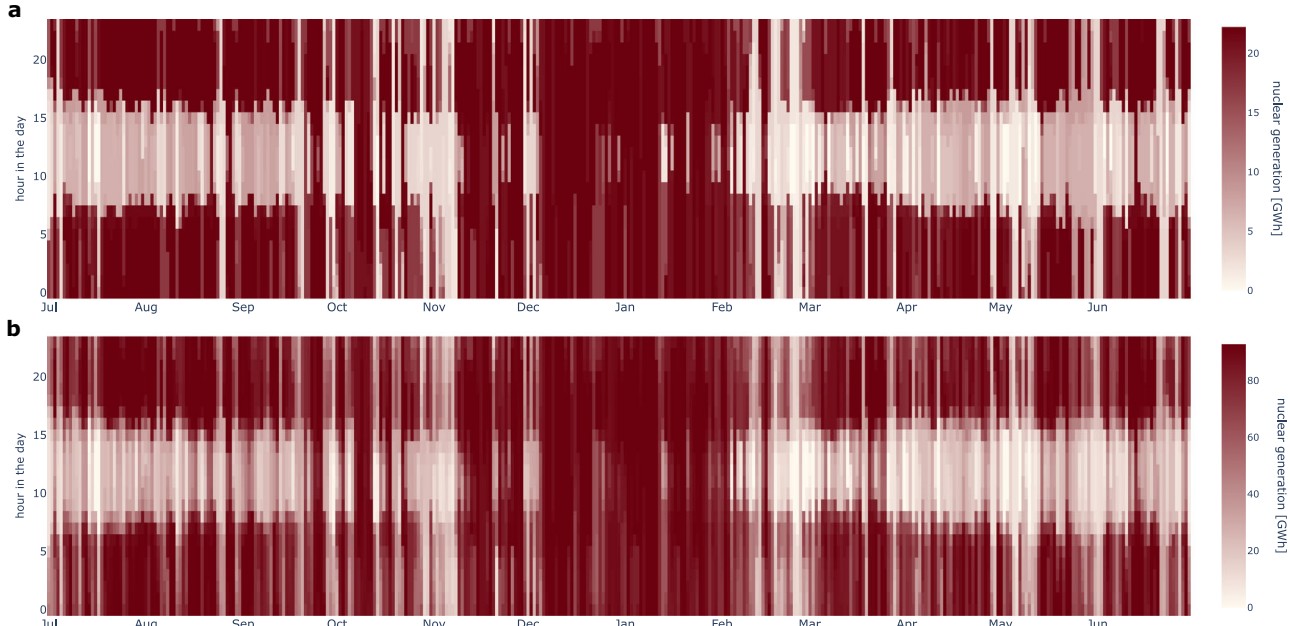

**Fig. 7 | Hourly and daily generation patterns of nuclear power aggregated across all countries in scenario (3) with policy-oriented exchange of electricity and hydrogen for 1996/97.** Supplementary Fig. 13 and Supplementary Fig. 14 show the operational patterns for all interconnection scenarios for low and high levels of nuclear power, respectively. **a** Low levels of nuclear power. **b** High levels of nuclear power.

electrolysis, which limits the need for capital-intense electrolysis capacity.

While long-duration storage typically provides most of the energy needed to cope with a renewable drought, it may not necessarily provide all the power capacity. To reduce the build-out of capital-intense hydrogen gas turbines for covering peak residual load events, battery storage is sometimes used in our least-cost system. Despite the additional energy losses from conversion, this may lead to charging and discharging cycles of batteries or pumped hydro while hydrogen gas turbines generate electricity (e.g., on December 12, 16, and 17 in Fig. 6b).

Short-duration flexibility options are also used in our model to balance brief events of renewable surplus generation within longer drought events. In such cases, the discharging of long-duration storage is interrupted, and batteries and pumped hydro storage are used instead for balancing (e.g., visible on December 16, 22, 25, or 26 in Fig. 6b). Here, the model favors shorter-duration storage over hydrogen-based storage because of lower roundtrip energy losses, indicating a "merit order" of storage technologies.

Finally, long-duration storage may also be used for diurnal cycling in summer in the least-cost system (Fig. 6c). Here, hydrogen-based storage integrates solar PV surplus energy, which reduces the need for short-duration battery capacity that otherwise would be required for this purpose. Importantly, the least-cost long-duration storage capacity is not defined by these diurnal variability patterns in summer but rather by major winter droughts. In other words, the long-duration storage capacity that is built to cope with winter droughts can have repercussions on the least-cost battery capacity and dispatch decisions for dealing with summertime solar PV variability.

These model-based findings complement those of a previous study focused on the US context, which demonstrated that long-duration storage can also provide short-term flexibility[91].

## Impact of firm zero-emission generation
In our default scenario, we allow for firm and variable renewable technologies only. However, nuclear power is considered a valid decarbonization option in some European countries. To reflect such country-specific energy policy strategies, we introduce two complementary scenarios with exogenous low or high levels of nuclear power, amounting to 24 gigawatts (GW) or 102 GW, respectively. We assume that nuclear power serves as a firm zero-emission generation capacity that does not face inter-hourly operational constraints but minor costs for ramping up or down that relate to wear and tear costs and potentially also to energy losses. In our model, nuclear electricity generation complements particularly diurnal solar PV variations and is operated at full capacity during extended periods in winter during extreme renewable portfolio droughts, which reduces the need for long-duration storage discharging. This is visible in the least-cost operational patterns of both low and high levels of nuclear power in the extreme winter of 1996/97 (Fig. 7), in which many countries are simultaneously affected by severe renewable droughts (Fig. 5). These operational patterns generally apply across all investigated weather years (Supplementary Fig. 15 and Supplementary Fig. 16). Additionally, the exogenous firm generation capacity generally decreases least-cost investments in VRE technologies (Supplementary Fig. 9, Supplementary Fig. 10, Supplementary Fig. 11), which mitigates the overall system's need for flexibility and long-duration storage.

Low levels of nuclear power reduce the least-cost long-duration storage energy capacity across all investigated interconnection scenarios only to a minor extent (Fig. 8). Without geographical balancing in scenario (1), low nuclear capacities mitigate least-cost long-duration storage investments only in those countries where nuclear power is deployed, decreasing the aggregated median (maximum) long-duration storage energy capacity by 4 (6)%. With policy-oriented cross-border exchange in scenarios (2) and (3), the mitigating effects of low levels of nuclear power on the aggregated median (maximum) storage energy capacity are slightly larger with 6 (7)% or 8 (7)%, respectively. This is because the firm generation technology mitigates renewable droughts both domestically and in other countries. Under the assumption of unconstrained geographical balancing in scenario (4), median long-duration storage energy reduces by 5%, while the maximum storage energy even slightly increases by less than 1% because of interactions with bioenergy in the least-cost system. Long-duration storage further remains necessary to continuously meet the

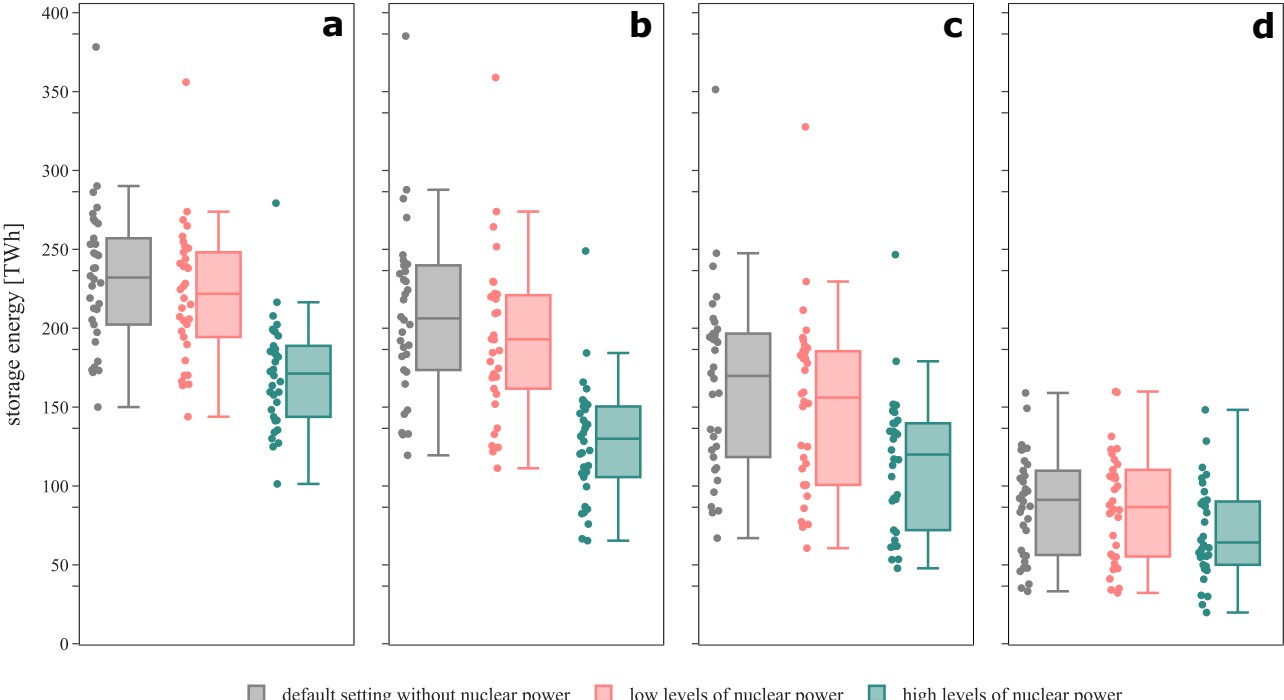

**Fig. 8 | Least-cost long-duration storage energy capacity aggregated across all countries for all modeled weather years and interconnection scenarios for different levels of nuclear capacities.** Each dot refers to one independently modeled weather year. The center line denotes the median, box limits indicate the interquartile range (Q1-Q3), whiskers extend to 1.5 × the interquartile range, and points beyond the whiskers represent outliers. **a** Scenario (1): no exchange of electricity nor hydrogen (island systems). **b** Scenario (2): policy-oriented exchange of electricity, no exchange of hydrogen. **c** Scenario (3): policy-oriented exchange of electricity and hydrogen. **d** Scenario (4): pan-European copperplate assuming unconstrained exchange of electricity and hydrogen.

flat hydrogen demand from coupled sectors, as assumed in our model (see Section "Formal definition of the hydrogen module" in the Supplementary Information).

In the scenario with high levels of nuclear power, higher shares of demand during extreme droughts can be met by nuclear, which reduces least-cost long-duration storage investments in our model-based analysis. This effect is complemented by nuclear disproportionately displacing least-cost investments into wind and solar PV, which in turn further lowers long-duration storage needs. Compared to the default scenario without nuclear power, these effects result in a substantial decrease of the median (maximum) storage energy capacity by 26 (26)% in scenario (1). Allowing cross-border electricity trade in scenario (2) enables nuclear power exports to balance renewable droughts domestically and in other countries, which further reduces long-duration storage energy capacity by 37 (35)% in our model. In scenario (3), the reduction of 29 (30)% is less pronounced because more renewable surplus is integrated, converted to electrolytic hydrogen, and shared across countries, leading to a more beneficial role of long-duration hydrogen storage. In scenario (4), nuclear power can balance renewable droughts across Europe without grid constraints, lowering least-cost mean storage energy by 30%. The maximum least-cost storage energy capacity declines by only 7%. One reason for this is that nuclear displaces bioenergy and some battery discharging capacity, resulting in a net firm capacity gain far less than the 78 GW additional nuclear capacity. Further, even during extended extreme droughts, renewable generation potential substantially above zero remains[33]. The substitution of wind and solar capacity by nuclear power thus yields a higher residual demand, which is met by firm technologies including long-duration storage. To conclude, our model-based analysis suggests that also in cases with high nuclear capacities, substantial long-duration storage investments remain optimal in the least-cost solution. In an additional sensitivity for Germany, we show that significant long-duration storage capacities remain part of the least-cost solution even in cases with much higher capacities of nuclear (or other firm zero-carbon technologies) exceeding by far the peak nuclear generation capacity ever realized in Germany (see Section "Additional information on the impact of firm zero-emission generation" in the Supplementary Information).

### Sensitivity analyses

Our analysis highlights the crucial role of long-duration storage for coping with pronounced renewable droughts in a climate-neutral European energy system based on renewable or zero-emission technologies. Alternatively, conventional fossil power plants combined with carbon dioxide ($CO_2$) removal could serve load during extreme drought events. A potential option for this is direct air carbon capture and storage (DACCS), which removes $CO_2$ emissions directly from the atmosphere and stores them underground without requiring proximity to emission sources[92]. In a sensitivity analysis of interconnection scenario (3) and weather year 1996/97, we allow for the endogenous deployment of oil-fired backup capacity and a corresponding use of DACCS. Unlike gas, oil can be easily stored above ground at low costs and transported via trucks, avoiding the need for extensive capital-intensive grid infrastructure. Since DACCS has not yet achieved maturity, its cost remains highly uncertain and depends on policy support, economies of scale, technological learning curves, and the profile of its electricity demand[92]. We account for these uncertainties by varying the $CO_2$ removal cost from 100 to 1,500 EUR per tonne (t) $CO_2$ (Fig. 9). The lower cost estimates in this range require a combination of optimistic assumptions, including continuous DACCS operation at low electricity price levels[92].

Only at very low DACCS costs of 100 EUR per t $CO_2$, 250 GW of oil-fired backup capacities are deployed in the least-cost system in Europe. These substitute 90% of the long-duration storage energy capacity and partially also wind, solar PV, batteries, electrolysis, and hydrogen turbines compared to our default scenario. However,

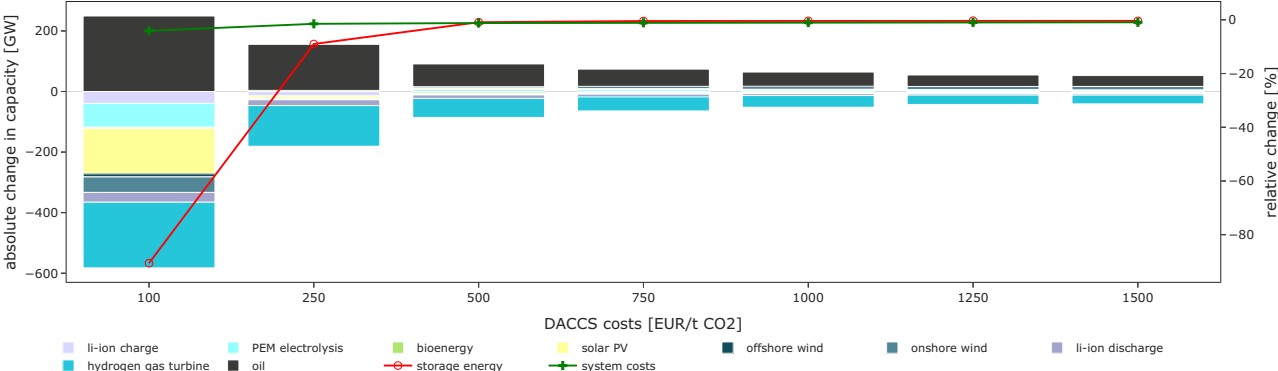

**Fig. 9 | Changes in least-cost capacities and system costs for varying costs of direct air capture and storage of emissions from the operation of oil-fired backup capacity.** Absolute change in generation and flexibility capacity (bars, left y-axis) and relative change of long-duration energy storage capacity and system costs (lines, right y-axis) aggregated across all countries in Europe in the interconnection scenario (3) compared to our default setting in 1996/97 for varying costs of direct air capture and storage of emissions from the operation of oil-fired backup capacity. For readability, we show the zero line of the left y-axis in gray.

significant energy storage capacity needs of over 30 TWh remain required in the least-cost solution. For DACCS costs above 100 EUR per t $CO_2$, the substitution effect for the long-duration storage discharging component diminishes substantially, while all other capacities are not affected. Likewise, only for very low levels of DACCS costs of 100 EUR per t $CO_2$, system costs reduce considerably by 4.0%. For all the other DACCS costs, system cost savings range only around 1.0%. That is, our results are very robust against introducing conventional electricity generation and DACCS, except if this option could be realized at extremely low costs. We find similar results for a related sensitivity that introduces load shedding, drawing on a wide range of different values of lost load (see Section "Additional information on a sensitivity with varying values of lost load" in the Supplementary Information).

The costs of onshore and offshore wind, as well as solar PV, have declined continuously over the last decades. They are projected to decrease further, although uncertainties remain concerning possible economies of scale and technological progress[6,93,94]. Similarly, hydrogen-based underground storage has so far been hardly deployed, also resulting in cost uncertainty. To address this, we analyze the sensitivity of least-cost deployment to variations in overnight investment costs of these technologies (Supplementary Table 1) for the policy-relevant interconnection scenario (3) and the year 1996/97, which contains the most extreme renewable drought in our data. The energy capacity costs of underground hydrogen storage are orders of magnitude lower than those of wind and solar PV. To reflect these cost asymmetries, we vary storage energy costs across a broader range, with a focus on high-cost scenarios (Fig. 10).

In our model, lower VRE or higher storage energy costs lead to overbuilding wind and PV to reduce reliance on storage. Although the latter reduces the potential to shift renewable surplus energy to periods with low VRE availability or high demand, the additional renewable capacity decreases residual demand that otherwise would require long-duration storage discharge. Conversely, lower storage energy or higher VRE costs favor long-duration storage deployment over VRE capacity expansion. This substitution particularly affects onshore wind. These patterns generally apply across all investigated cost scenarios. Moderate deviations from the default setting of future wind and solar PV costs appear most relevant[6]. For these scenarios, least-cost storage deployment is hardly affected in our parameterization.

## Discussion

This paper analyzes how extreme renewable energy droughts impact least-cost investments and operations of long-duration electricity storage in a future renewable European power sector by combining renewable availability time series analysis for drought identification and energy system modeling.

Our analysis yields several key insights. Extreme renewable energy droughts identified for the policy-oriented renewable capacity portfolios simulated here, which may last several weeks or even months, define the least-cost operational and investment needs for long-duration storage determined in our model. Firm renewable energy sources, such as hydro reservoirs or bioenergy reduce least-cost long-duration storage capacities for dealing with extreme droughts, while co-occurring high-demand periods exacerbate them. Our model-based findings highlight that long-duration storage is an indispensable technology for coping with very pronounced renewable droughts in fully renewable energy systems with significant power supply and demand seasonality.

Interconnection among European countries can reduce least-cost long-duration energy storage capacity. While previous literature has found a similar effect in case studies covering parts of the US or Europe and a limited selection of weather years[32,95-97], we show that this finding also holds for the pan-European renewable energy system including 33 countries optimized here, particularly considering the most pronounced simulated drought events that occur in a large dataset of 35 weather years. Yet, the storage-mitigating effect of policy-relevant interconnection levels of the TYNDP 2022 remains limited and could only be substantially reduced in scenarios with interconnection levels far beyond envisaged grid expansion plans. Our analysis shows that additional cross-border exchange of renewable hydrogen, as planned in the European Union, contributes to reducing least-cost long-duration storage needs.

Our model results suggest that there is a sizable need for long-duration storage in a fully renewable European power sector, irrespective of the extent of the assumed interconnection. Even in a perfectly interconnected Europe, which allows for unconstrained spatial smoothing of the most extreme drought in the data, 159 TWh of long-duration storage energy capacity result in the least-cost solution of our baseline parameterization, corresponding to around 3% of yearly electric load. This number indicates that there is a lower bound for storage energy capacity required in a fully renewable European energy system. Its magnitude appears sizable, particularly considering that barely any hydrogen-based long-duration storage capacity is currently installed in Europe[98]. In more policy-relevant TYNDP interconnection scenarios between European countries, the least-cost long-duration storage investments substantially increase in our model to 351 TWh, or more than 7% of yearly European electricity demand. This finding leads to the conclusion that transitioning to a renewable European energy system may require significant long-duration storage capacities in the

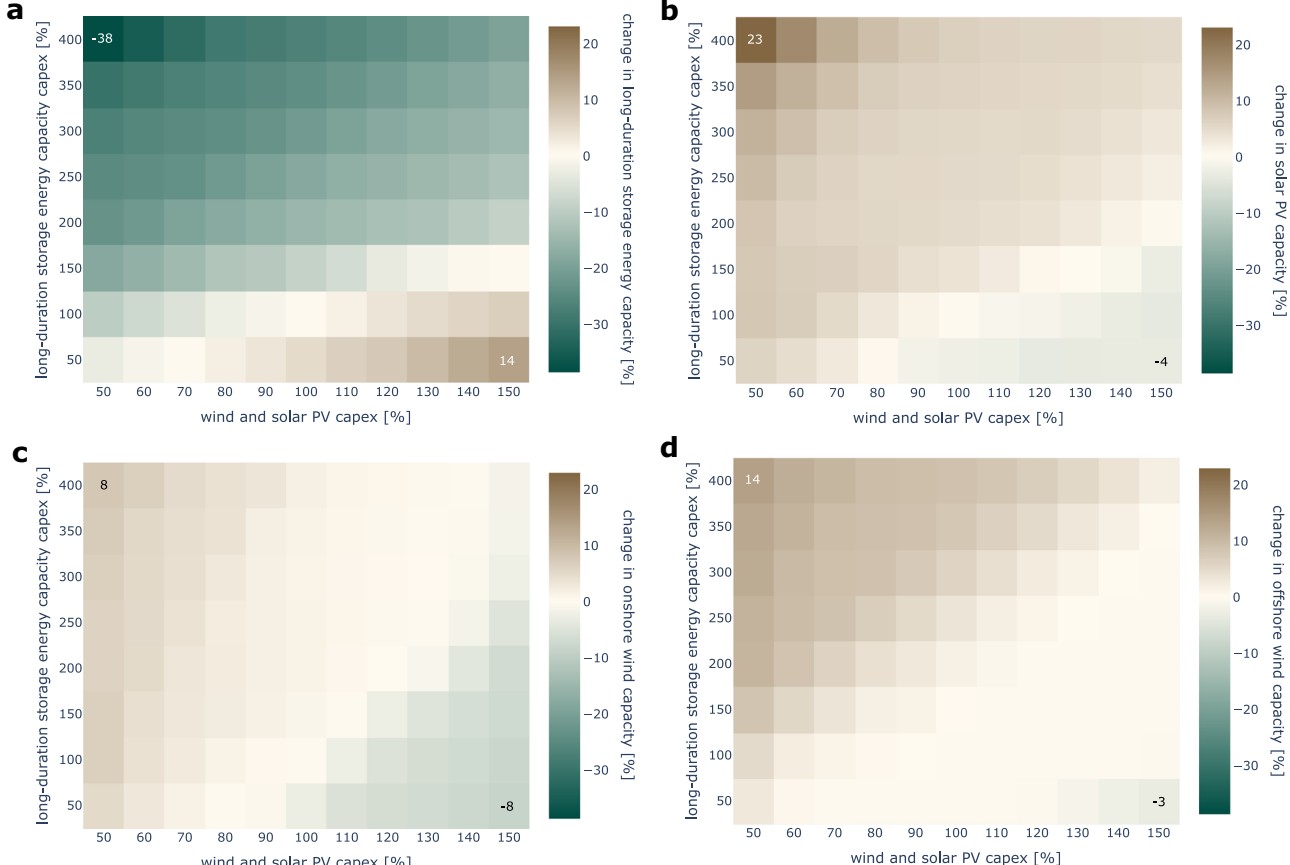

**Fig. 10 | Relative changes in least-cost investment decisions for varying levels of investment costs of onshore wind, offshore wind, solar PV, and long-duration storage energy capacity in 1996/97 aggregated across all countries.** Cost and deployment variations are denoted in percentage. A cost variation of 100% refers to the original parameterization. The annotated scenarios refer to the most pronounced changes across all scenarios. **a** Change in long-duration storage energy. **b** Change in solar PV capacity. **c** Change in onshore wind capacity. **d** Change in offshore wind capacity.

order of several hundred TWh, particularly for coping with extreme, yet rarely occurring, Dunkelflaute events. Our results extend previous analyses that focus on droughts in single countries, such as Germany, where long-duration storage capacities of 9%[68] or 10%[67] of annual electricity demand are required in a least-cost system configuration for dealing with extreme drought events. The slight differences in storage needs are likely driven by the storage-mitigating effect of geographical balancing across countries, which we include in our modeling.

Overall, realizing the energy capacity levels determined by our model appears technically feasible, considering that around 1700 TWh of natural gas storage is currently installed in Europe[99], even when taking into account the lower volumetric energy density of hydrogen compared to methane. Yet, experience with converting existing gas storage to hydrogen or building new underground hydrogen storage is so far limited[98]. The same applies to hydrogen grids as well as hydrogen generation and re-conversion infrastructure.

Importantly, large-scale adoption of hydrogen-based long-duration storage, comprising electrolyzers, caverns, and hydrogen turbines for reconversion, will likely have long lead times because of supply chain and permitting bottlenecks[100,101]. To safeguard the renewable energy transition in Europe, system planners and policymakers should thus consider early action to enable rapid scaling for realizing hydrogen storage investments. Furthermore, the maximum long-duration storage energy capacity in Europe identified in the least-cost system configuration, driven by the most extreme renewable drought in the winter of 1996/97, exceeds the next highest storage need determined for the weather year 1984/85 by 42%. Market actors are unlikely to invest in such rarely utilized long-duration storage capacity without

additional deployment incentives. Therefore, targeted support instruments or capacity mechanisms may be necessary to ensure the realization of sufficient storage capacity.

We also observe complex interactions of long-duration storage with short-duration batteries and other flexibility options. A complementary operation of these technologies can mitigate least-cost long-duration storage and battery charging and discharging capacities to some extent, hence decreasing total system costs.

Our findings prove robust against the addition of moderate levels of flexible nuclear generation, which mitigate the least-cost long-duration storage investments in our model only to a minimal extent. In a setting with substantially higher nuclear generation capacity, the storage-mitigating effect is more pronounced. Yet, substantial long-duration storage investments remain optimal in the least-cost system configuration. As we largely abstract from inter-hourly flexibility constraints, nuclear's potential to substitute long-duration storage needs as found here can be interpreted as an upper bound of this storage-mitigating effect. Including more specific operational constraints would likely reduce it. Notably, the role of nuclear in a climate-neutral European energy system is contentious due to scalability challenges, exceptionally long construction times, and high uncertainty in final investment costs as evident in ongoing expansion projects in Europe and globally[102]. Likewise, nuclear waste management faces substantial economic, environmental, and societal challenges. Additionally, the future deployment of other advanced firm zero-emission generation technologies remains highly uncertain, as these require very optimistic cost assumptions to be competitive against variable renewables in combination with long-duration storage[103].

A sensitivity analysis for the weather year 1996/97, which includes the most extreme simulated renewable drought in the data, shows that the least-cost deployment of fossil backup capacity in combination with carbon abatement via DACCS could result in minor system cost decreases, but only if DACCS costs are extremely low. For higher DACCS costs, which appear more likely given current knowledge[92], the system cost effect is only marginal. Under these scenarios, such backup capacity would only proportionally displace long-duration storage discharge capacity while storage energy capacity remains hardly affected in the least-cost system. Considering the significant uncertainty concerning realizable DACCS costs and implementation levels, our analysis suggests that the role of fossil fuel-based backup capacity with emission abatement for mitigating long-duration storage remains limited in a climate-neutral European energy system.

Next, our analysis shows that the selection of weather years has a notable impact on least-cost long-duration storage use, which confirms previous research[34,36,82,87]. Different weather years result in widely varying least-cost long-duration storage investments, particularly in scenarios with constrained interconnection. However, due to computational limitations, many policy-relevant studies rely on only one or a limited set of weather years. For instance, the TYNDP 2022[18], which is a crucial European planning tool to determine cross-border infrastructure needs, draws on three weather years (1995, 2008, and 2009) for its long-term scenarios for 2050. Our analysis shows that the least-cost long-duration storage capacity considering TYNDP interconnection levels in the corresponding summer-to-summer weather years (1994/95, 1995/96, 2007/08, 2008/09, and 2009/10) ranges from 103 to 239 TWh. Notably, we find the least-cost storage capacity required to balance the most extreme drought in winter 1996/97 is 47% higher than this range's upper bound. This underscores the importance of considering multiple weather years for identifying weather-resilient system configurations and long-duration storage sizing, particularly those that include the most pronounced drought events.

For Europe, the power sector impacts of VRE droughts are most pronounced in winter, particularly in the winter of 1996/97[33]. We thus argue that our summer-to-summer modeling approach better captures the effects of compound drought events spanning across the turn of years, as compared to models using single calendar years.

Finally, there is no consensus on the definition of variable renewable energy droughts[22]. In the energy policy debate, events lasting from just a few hours to one or two weeks have been labeled as Dunkelflaute[13,19,20,104]. Based on our analysis, we propose refining the Dunkelflaute notion to focus on events with the most significant implications for long-duration power sector flexibility: extended (winter) periods where renewable energy falls short of electricity demand, which ultimately define the energy capacity and the operation of long-duration storage. We suggest not using the term Dunkelflaute for very short periods of low wind and solar availability, especially not for a few hours within a day.

Like in any model-based analysis, we made several assumptions and simplifications, whose qualitative effects we discuss briefly in the following. First, space heating will likely be electrified to a substantial extent in the future in Europe[105]. This will not only increase annual electricity demand but also lead to a more pronounced demand seasonality with higher load in winter. While we account for the projected additional electricity demand via linear scaling, our demand time series, which we chose to ensure consistency with the renewable energy time series, only have moderately seasonal profiles as projected for 2025 (see Section "Input data, technology portfolio, and capacity bounds" in the Supplementary Information). Alternative demand profiles with more pronounced heating-related seasonality are likely to cause higher least-cost long-duration electricity storage investments than modeled here. However, these could in turn be mitigated by long-duration thermal energy storage[106,106], which we abstract from here. Similarly, winter heat demand peaks could be addressed by building

insolations[107]. We expect that these factors will balance each other to some degree with respect to the needed long-duration electricity storage capacities. Yet, analyzing the interactions of different types of thermal storage, heating technologies, and building renovation with the power sector in detail is a complex task[107,108], requires a dedicated research design beyond the scope of this paper, and merits future research.

Similarly, electricity demand is likely to increase in summer in many European countries because of a growing demand for cooling[109]. While this may increase flexibility requirements of the future energy system, we expect the additional demand from cooling would affect short-duration flexibility options rather than long-duration storage, considering typical diurnal cooling patterns and their partial coincidence with solar PV generation peaks. Our analysis also indicates that the most pronounced renewable energy droughts, which also drive long-duration storage needs, primarily occur in winter and not in summer in an interconnected European energy system.

Next, our analysis excludes industrial and commercial load shifting or shedding as well as optimized grid interactions of battery-electric vehicles. While these flexibility options would generally enhance system flexibility, their impact on long-duration storage capacity remains likely limited because of their short duration. Similarly, we do not consider long-duration storage technologies other than hydrogen cavern and porous storage in combination with electrolyzers and hydrogen turbines. For example, methanol-based long-duration storage in combination with Allam cycle turbines could be a potential alternative[34]. This technology does not depend on highly localized underground sites but allows for aboveground storage that can generally be placed anywhere[98,110], which could lead to spatial redistribution of least-cost storage capacity. Further, if this alternative storage technology came with higher roundtrip efficiencies or with lower costs than assumed here for hydrogen-based long-duration storage, least-cost storage capacities aggregated across all countries are likely to increase compared to our results. Importantly, we do not argue that hydrogen-based underground storage in combination with electrolyzers and hydrogen turbines is the only or best technology to serve as long-duration storage. We have chosen this technology as it seems the very promising given current knowledge. The least-cost long-duration storage capacities found in this analysis can also be considered as the energy amounts that need to be shifted or covered by any alternative technology.

Importantly, our analysis focuses exclusively on least-cost solutions and does not account for near-optimal system configurations, which could, for instance, exhibit lower implementation barriers or greater societal acceptability. Future work could employ a modeling-to-generate-alternatives approach to explore the extent to which such near-optimal capacity layouts exist and at what additional system costs[111]. Likewise, it would be of interest to investigate how uncertainties over future technology cost or demand developments impact storage outcomes, e.g., by using Monte Carlo simulations or other techniques for optimization under uncertainty.

Further, the model used in this analysis is deterministic and assumes perfect foresight. Under limited foresight, long-duration storage might be operated more precautiously compared to our analysis, i.e., stockpiling more energy over longer time spans and avoiding very low states of charges to hedge against extreme drought events that may occur later in planning horizon[106]. Accordingly, the least-cost operational and capacity decisions of the long-duration storage found in this analysis should be interpreted as benchmark solutions. Under imperfect foresight, which applies to operators in the real world, long-duration storage needs may exceed those determined here. Likewise, energy system models that incorporate stochasticity may reach different conclusions regarding the role of technologies dominated by operational costs, such as DACCS, warranting future research.

Moreover, we do not account for hydropower droughts, neither in the time series analysis nor in the capacity expansion model. Analyzing hydropower droughts and their interactions with wind and solar droughts appears to be a promising avenue for future research. At the same time, it appears unlikely that including hydro droughts would qualitatively alter our main findings, as the number of affected countries and the overall contribution of hydro power to European electricity supply remains limited.

Future work could investigate the effects of different time horizons for energy modeling, ranging from single years to several years[36], yet with a particular focus on renewable drought events. This also includes quantifying the distortions of modeling either summer-to-summer periods, as done in our study, or winter-to-winter periods. And while perfect foresight within a year results in relevant benchmark outcomes, considering imperfect foresight for long-duration storage operations within a year may offer complementary insights into real-world storage use. Further, exploring the impact of future changes in renewable energy droughts and storage needs driven by climate change appears desirable. This could build on an emerging body of research on climate-driven changes in wind and solar generation and datasets[112,113].

## Methods

In this study, we combine two methods: a VRE drought analysis based on availability time series of wind and solar power, and a cost-minimizing capacity expansion model of the European power sector.

### Method and data for identification of variable renewable energy droughts

We use the open-source tool variable renewable energy drought analyzer (VREDA) to identify and evaluate VRE drought patterns based on availability time series. VREDA has been designed to implement good practices of multi-threshold drought identification[22] and has been applied before to characterize drought patterns in Europe[33]. It employs the varible-duration mean-below-threshold (VMBT) method for drought identification, which varies the permissible drought duration between two full calendar years and one hour. This method searches for periods where renewable availability has a moving average below a specific drought qualification threshold by iteratively decreasing the drought duration. In each iteration, the algorithm sets the averaging interval to the respective event duration. Initially, it searches for drought events that last two full years and iteratively decreases the averaging interval to identify shorter events. A time series section with a moving average below the drought threshold identifies a drought event. It is then excluded from subsequent iterations, in which the averaging interval decreases further and additional (shorter) events are identified.

The iterative procedure of VMBT overcomes shortcomings of previous research[22]. The method allows for pooling of adjacent periods that independently may not qualify as VRE drought to capture longer-lasting events, i.e., intermediate periods with a renewable availability above the drought threshold. It further identifies unique events, avoids double counting as well as overlaps with adjacent events, and captures the full temporal extent of drought periods ("event" definition).

We use country-level VRE availability time series provided by the Pan-European Climate Database, including 35 weather years from 1982 to 2016 for on- and offshore wind as well as solar PV[114]. Using a summer-to-summer planning horizon, this means that we cover 34 complete winter seasons.

For comparability across regions, drought thresholds are scaled relative to specific fractions of country-specific long-run mean availability factors for the period 1982 to 2016[22]. These fractions range from 10 to 100% of mean availability, increasing in 5% increments and reflecting different levels of drought severity. We refer to these fractions to as "relative thresholds". Periods that qualify as drought events based on lower thresholds are likely brief and severe with very low availability. In contrast, events identified by higher thresholds may last substantially longer, i.e., up to several weeks or months[33].

We analyze drought patterns for renewable technology portfolios, comprising on- and offshore wind power as well as solar PV, for two cases that differ regarding the assumed electricity transmission between countries[22]: completely isolated countries ("energy islands") or perfect interconnection across all countries (pan-European "copperplate"). For the energy islands scenario, we combine all technology-specific time series into a portfolio time series using capacity-weighted averages. The respective weights are based on policy-relevant assumptions on renewable capacity mixes from the TYNDP 2022 "Distributed Energy" scenario (default) and "Global ambition" scenario in a sensitivity analysis[115]. We update these assumptions for Germany according to the latest government targets[116]. For the copperplate scenario, we combine all country-level portfolio time series into a single pan-European composite time series, using weights according to the TYNDP 2022 (scenario "Distributed Energy").

We use the "drought mass" metric[33] to identify extreme drought events by aggregating the drought patterns of numerous single-threshold analyses, ranging in 5%-increments from the 10% to 75% threshold. To compute the drought mass score, we first modify these patterns by assigning the value 1 to drought hours and the value 0 to those hours that do not qualify as drought for each threshold. Next, we equally weigh the resulting drought patterns across all thresholds. We then accumulate the hourly scores up to the cut-off threshold 75%, excluding the drought patterns based on higher thresholds. This approach determines the multi-threshold event duration according to the 75%-analysis, while the event drought mass aggregates the drought patterns identified by included thresholds. The highest cumulative score identifies the most extreme event per summer-to-summer planning horizon.

Wind droughts are more frequent and severe in summer than in winter[33,44,47]. In countries with high wind shares in their capacity portfolio, the most extreme renewable portfolio drought events may thus occur in summer. Since peak electricity demand periods usually occur in winter in Central and Northern European countries, summer droughts generally matter less than winter droughts. To account for this, we compute a yearly drought mass score for droughts occurring throughout the summer-to-summer planning horizon and a winter drought mass score for drought events between October and March. When illustrating the relation of drought patterns and long-duration storage operation, we display both the most extreme summer and winter droughts if the highest drought mass score relates to summer drought (compare gray and teal boxes in Figs. 1 and 7). Conversely, we mark only one event if the highest yearly drought mass score refers to a winter drought (compare teal boxes only in Figs. 1 and 7).

### Power sector model

We use the open-source dispatch and capacity expansion model of the European power sector Dispatch and Investment Evaluation Tool with Endogenous Renewables (DIETER)[25,117] to analyze the interaction between VRE droughts and long-duration storage capacity investments in a fully renewable European power sector. The model features a simple transport model for exchanging electricity across countries, abstracting from grid constraints within countries. Different model versions have been applied to study various aspects of VRE integration and their interaction with flexibility options or sector coupling technologies[32,35,106,108,118–125]. DIETER is a linear program that determines least-cost capacity and dispatch decisions, optimizing over all contiguous hours of a full year under perfect foresight.

Exogenous model inputs entail techno-economic parameters, such as investment and variable costs, availability time series of wind and solar PV, as well as price-inelastic demand time series for electricity and hydrogen. Model results can be interpreted as the outcomes of a

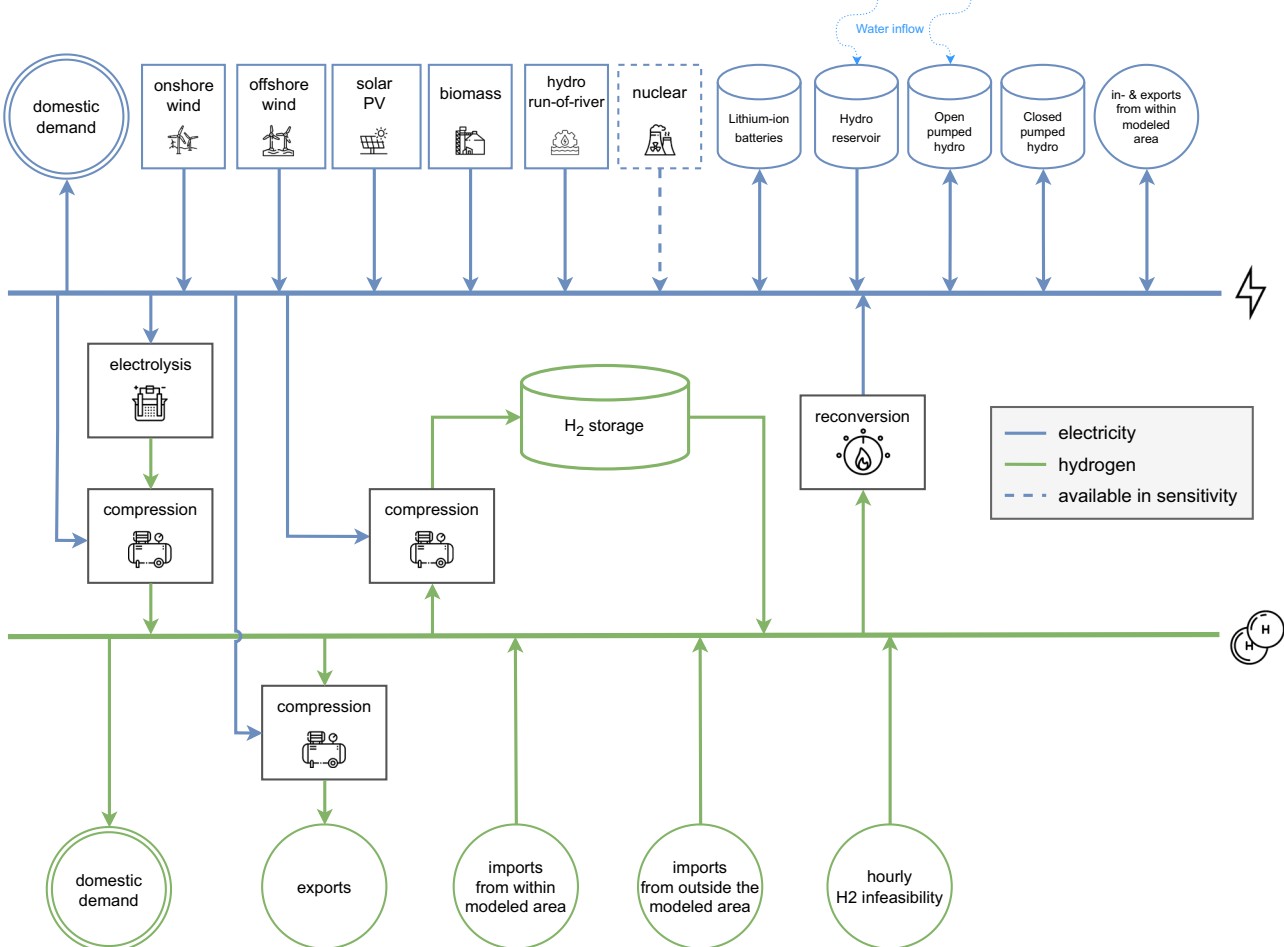

**Fig. 11 | Schematic overview of the model DIETER.** The figure depicts a schematic representation of the energy system model DIETER, which has been used in this study. Blue elements refer to electricity-related elements, to hydrogen-related components. They are connected by electrolysis and reconversion. This cover has been designed using resources from Flaticon.com.

perfect, frictionless European electricity market, where all power generators maximize their profits. Costs are minimized for the overall system, depending on interconnection assumptions. In the energy islands case, costs are minimized for every country in isolation.

Figure 11 provides an overview of the model's structure, how and with which technologies electricity and hydrogen can be generated, stored, and exchanged.

While the general model formulation of DIETER has been described extensively in the papers mentioned above, we use a version that features an improved representation of hydrogen technologies in this study. The model includes the generation, storage, and transport of renewable hydrogen technologies as well as its re-conversion to electricity. The hydrogen-based long-duration storage energy capacity denotes the lower heating value of the storage working gas. A formal definition of the additional equations is available in Section "Formal definition of the hydrogen module" in the Supplementary Information.

### Scenarios

Our analysis comprises 33 European countries (EU27 excl. Malta and Cyprus, the UK, Norway, Switzerland, and the Western Balkans). We analyze and compare the impact of VRE droughts on least-cost long-duration storage capacities for four different interconnection scenarios: (1) all countries operate as energy islands without exchange of electricity or hydrogen between countries; (2) cross-border exchange of electricity in line with the assumptions on the

European transmission grid in 2050 projected by the TYNDP 2022[115] (scenario Distributed Energy), while hydrogen exchange remains disabled; (3) cross-border electricity and hydrogen exchange according to the TYNDP 2022 (scenario Distributed Energy); and (4) unlimited exchange of electricity and hydrogen, i.e, all countries are perfectly integrated as pan-European copperplate. By design, the VRE drought analysis based on renewable availability time series does not account for limited cross-border exchange. Consequently, it is conducted only for the two counterfactual interconnection scenarios (1) and (4). In the latter case, the drought identification is based on a renewable availability time series that combines all countries into a capacity-weighted single pan-European composite time series (compare Section "Method and data for identification of variable renewable energy droughts").

These varying degrees of interconnection allow distinguishing several effects. First, scenario (1) identifies maximal long-duration storage needs across European countries, excluding geographical balancing of VRE droughts between countries[33]. Second, comparing scenarios (2) and (3) disentangles the effects of policy-oriented electricity and hydrogen interconnection levels on long-duration storage needs. Finally, the copperplate scenario (4) identifies minimal long-duration storage needs, including unconstrained geographical balancing of VRE droughts, which can be interpreted as unavoidable or "no-regret" investments in a fully renewable European energy system. We abstract from grid investment costs but impose losses on hydrogen

exchange related to compression for long-distance transport via pipeline.

We investigate a large number of weather years that differ in renewable availabilities and demand patterns. These weather years range from 1982 to 2016. Extreme VRE droughts often occur in European winter, may last up to several weeks, and span across the turn of years[33]. Hence, we use a summer-to-summer planning horizon (in line with refs. 67,72, but unlike, e.g., ref. 87), comprising 8760 consecutive hours. Note that we independently optimize the capacity mix for every year.

In two additional scenarios, we exogenously add different levels of nuclear power to the generation mix, aligned with the "Distributed Energy" and "Global Ambition" scenarios from the TYNDP 2022, respectively[115]. The first of these scenarios assumes low levels of nuclear capacity in Finland, France, Romania, Slovakia, and the UK, totaling 24 GW. The second scenario scenario assumes high levels of nuclear power, assuming additional deployment in the formerly mentioned countries as well as in Bulgaria, the Czech Republic, Hungary, Poland, Sweden, and Slovenia, totaling 102 GW.

### Input data, technology portfolio, and capacity bounds

To quantify the maximum impact of VRE droughts on long-duration storage, we model fully renewable supply scenarios in our default setting, excluding carbon removal and fossil fuel-based dispatchable generation technologies. This is not enforced by binding renewable or carbon emission targets[121], but rather by limiting the available generation technology portfolio to zero-emission options. These include solar PV, on- and offshore wind power, bioenergy, and different types of hydroelectric power (run-of-river, reservoir, pumped-hydro).

For policy relevance, we allow generation capacity expansion within lower and upper potentials of the TYNDP 2024[126]. Additionally, we assume an annual generation limit for bioenergy in line with the TYNDP 2022[18]. Installed power and energy capacities of the available hydro technologies are fixed to values provided in European Resource Adequacy Assessment (ERAA) 2021[127].

We include underground hydrogen storage as a long-duration storage option. Underground cavern and porous storage energy potentials vary substantially across countries, including newly built and retrofitted storage facilities from natural gas infrastructure[98,110]. We constrain the expansion of long-duration storage energy capacities accordingly. For simplicity, we abstract from any differentiation between underground storage types and aggregate their potentials for each modeled country.

In our analysis, we attempt to illustrate the impact of persistent VRE droughts lasting longer than a few days on the power sector, notably regarding long-duration electricity storage. We thus abstract from an explicit representation of sector coupling options as well as seasonal heat storage. To this end, we use near-term demand profiles, retrieved from ERAA 2021[127] (representative for the year 2025). The profiles are scaled to the annual demand levels of the TYNDP (scenario Distributed Energy for 2050) and are reduced by the electrical energy amount required for the generation of the exogenous hydrogen demand.

We use the Pan-European Climate Database for renewable availability factors of on- and offshore wind and solar PV[114], as well as the hydro inflow, retrieved from the ERAA 2021[127].

### Data availability

The input data analyzed by the VRE drought analysis tool are a refined version of the Pan-European Climate Database (PECD 2021.3) and publicly available at Zenodo[114]. The input data of the capacity expansion model of the European power sector are also publicly available on GitLab at https://gitlab.com/diw-evu/projects/power-sector-droughts/. The power sector outcomes generated in this study have been deposited in a publicly available Zenodo repository[128].

### Code availability

For transparency and reproducibility, we provide the code of the VRE drought analysis tool in public repositories under permissive licenses available on GitLab at https://gitlab.com/diw-evu/variable_renewable_energy_droughts_analyzer. The code and manual of the capacity expansion model of the European power sector are also publicly available on GitLab at https://gitlab.com/diw-evu/projects/power-sector-droughts/.

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

## Acknowledgements

We thank the entire research group "Transformation of the Energy Economy" at the German Institute for Economic Research (DIW Berlin) for valuable inputs and discussions, as well as conference participants of the International Conference Energy & Meteorology 2023, the International Association for Energy Economics Con-ference 2023, the ENERDAY Conference 2024, and the Annual Conference of the European Association of Environmental and Resource Economists 2024 for valuable comments on earlier drafts. The authors acknowledge support for the research of this work from the Einstein Foundation Berlin (grant no. A-2020-612) and by the German Federal Ministry of Education and Research via the "Ariadne" projects (Fkz 03SFK5NO & 03SFK5NO-2).

## Author contributions

M.K. (lead), A.R. (support), and W.S. (support) developed the concept of the study. M.K. developed the methodology and the time series analysis tool. M.K. (lead), A.R. (support), and W.S. (support) developed the energy system model. M.K. (lead) and A.R. (support) were responsible for soft-ware development and usage. M.K. (lead), A.R. (support), W.S. (support) carried out the investigation. M.K. and A.R. curated data requirements. M.K. compiled visualizations. M.K. (lead), A.R. (lead), W.S. (support) wrote the original draft. M.K. (lead), A.R. (support), and W.S. (support) reviewed and edited the manuscript. W.S. was responsible for project acquisition and administration.

## Funding

## Competing interests

The authors declare no competing interests.
