## [Transparent Peer Review file · Nature Communications]

Long-duration electricity storage needs for coping with Dunkelflaute events in Europe

Corresponding Author: Dr Martin Kittel

Version 0:

Reviewer comments:

Reviewer #1

(Remarks to the Author)

This is a novel and valuable piece of work on the management of renewable energy droughts, especially considering the increasing need for large-scale deployment of weather-dependent renewables in decarbonized power systems as a core element of climate mitigation strategies. Unlike many previous studies, this research offers three key contributions

- Application of a New Drought Mass Metric: This study applies a new drought mass metric by integrating the Variable-Duration Mean-Below-Threshold (VDMT) method with multiple drought thresholds. Although the approach is based on the authors' previous work, it is more comprehensive than most other studies on the same topic, which typically rely on a single arbitrary threshold for drought identification.
- Combining Drought Identification with Energy System Modeling: Unlike most prior work that analyzes droughts solely from the supply side of VRE generation and in isolation, this paper combines drought identification methods with energy system modeling. This integration allows for an in-depth analysis of long-term (hydrogen-based) storage requirements and their operation under weather year variability within an energy system context.
- Assessment of Different Grid Configurations: The study considers various grid configurations, including autarky, copperplate, and limited cross-border capacity. This enables an examination of the effects of virtual storage, which can help mitigate the impacts of renewable energy droughts.

However, the paper also presents a few shortcomings or arguable methodological choices, particularly related to the selection of assumptions, which should be addressed or further clarified by the authors

- The authors use seasonal-agnostic thresholds (i.e., based on fractions of the long-term mean capacity factors) to identify droughts. However, this approach does not account for seasonality and, therefore, cannot distinguish between low production events caused by regular seasonal dips and those driven by true weather anomalies or extremes. Employing seasonal thresholds is arguably crucial to improve the efficiency of identifying potentially worst weather years that contain the most extreme droughts, which are critical for power system operations. Ultimately, the primary motivation for drought identification is to enhance the efficiency of weather year selection for energy system modeling and, consequently, to support the planning of a more resilient energy system—especially when dealing with large samples of weather year data.
- The installed capacity allocation of VRE assets/technologies within and across countries is used as a weighting factor to derive country-wide generation profiles (as in the autarky case) and EU-level generation profiles (as in the copperplate case). These generation profiles are subsequently used for drought identification, capacity expansion, and impact assessment. However, the capacity allocation is primarily based on a single scenario—the TYNDP 2022 'Distributed Energy' scenario—which may affect the robustness of the quantified results. It is recommended to incorporate capacity allocations from alternative sources to assess the sensitivity of the results.
- The near-term demand profiles are adopted but scaled to the 2050 demand levels based on the TYNDP 'Distributed Energy' scenario. However, the 2050 demand levels also account for the electrification of the space heating and cooling sectors, which would inevitably result in a different demand profile compared to the near-term one. This approach may introduce consistency issues, and the authors should clarify the underlying rationale.

(Remarks on code availability)

Reviewer #2

(Remarks to the Author)

This paper examines the need for long-duration energy storage in the context of a wind and solar reliant European electricity system. While I could quibble about some of the choices made, overall, the analysis is performed at a high level of quality. The paper is well written and organized. The results are consistent with a large body of work.

The main question I have about this study is the degree to which it is novel. The main concern about how they express their findings is the degree to which they appear to make claims about the real world based on a limited number of model simulations under a specific set of assumptions.

This paper concludes that several weeks worth of long-duration storage are needed to cost-effectively provide a reliable electricity system that is reliant on wind and solar generation. This conclusion is consistent with a large number of other studies reaching similar conclusions (e.g., Albertus et al., 2020; [https://www.cell.com/joule/fulltext/S2542-4351\(19\)30539-2](https://www.cell.com/joule/fulltext/S2542-4351(19)30539-2)).

They also conclude that there is a tradeoff between long-duration storage requirements and the geographic integration of electricity grids. This conclusion also has been reached by a number of previous studies (e.g., Staadecker et al., 2024; <https://www.nature.com/articles/s41467-024-53274-6>).

They also note that long-duration storage technologies can also meet short-duration storage needs. This conclusion too has been reached in other studies (e.g., Li et al., 2024; <https://pubs.acs.org/doi/full/10.1021/acs.est.3c10188>)

This paper seems to be professionally done and adds incrementally to our understanding. The main thing that is a little disconcerting about this paper is the degree to which it makes very strong and specific claims based on a very limited number of stylized simulations, while appearing to claim generality.

Specifically, costs of various technologies are likely to vary in the future, with many of them decreasing (e.g., Hunter et al., 2021; [https://www.cell.com/joule/fulltext/S2542-4351\(21\)00306-8](https://www.cell.com/joule/fulltext/S2542-4351(21)00306-8)). The specific amounts of long-duration storage in their optimizations will be sensitive to cost assumptions, yet there are no sensitivity studies looking at how results might be sensitive to cost. Long-duration storage is used only because it is cheaper than building more capacity, so the amount of long-duration storage “needed” is a function of the costs of the various technologies and not a generalizable fact.

In this light, the authors make a very quick jump from their model simulations to the real world, and talk as if their model results apply directly to the real world. Whenever they give numerical results they should be clear they are talking about their model, and not make the assumption that their model results directly apply to the real world. The authors are aware of the simplifications and assumptions inherent in their analysis, and thus should be cautious about stating specific quantitative results as they apply to the real world.

An example of this tonal issue can be seen in the abstract, where they write:

“Assuming policy-relevant interconnection, long-duration storage of 351 TWh or 7% of yearly electricity demand is required to deal with this event.”

What they mean to say is:

“Assuming policy-relevant interconnection, in the model configuration used here (including our specific cost assumptions, etc), the least-cost solution deploys 351 TWh (or 7% of yearly electricity demand) of long-duration storage to deal with this event.”

I do not see any discussion of value of lost load or unmet demand. What are the maximum dual values across all time steps and nodes? This would indicate the shadow cost of electricity in a perfectly efficient market. In many optimizations, this value can be extraordinarily high. Do the model results vary substantially if a “reasonable” value of lost load is used?

They say “In a sensitivity analysis, we add a generic firm zero-emission generation technology.” Very clearly, if this technology were very low cost, no long-duration storage would be needed. If it were very costly, then introduction of this generation technology would have no effect on solutions. I don’t know how hard it is to run the model, but it would be interesting to sweep through costs of this generic dispatchable generator and see how long-duration storage needs vary with the cost of this generator.

Further, often the potential surface that defines the optimal solution is very broad, and technology mixes that are far from the optimal mix might increase system cost only modestly. Given the large number of uncertainties regarding cost, permitting, demand, etc., it is useful to think about near-optimal solutions (c.f, Neumann and Brown, 2021; <https://www.sciencedirect.com/science/article/pii/S0378779620304934>).

Minor points:

It would be useful to have a schematic diagram showing what energy technologies are considered in the model. The schematic wiring diagram could represent the entire electricity grid as a single node, and the technologies considered by different boxes. This would let the interested reader quickly see which technologies are being considered.

This sentence needs some expansion (page 6):

“Note that VRE drought analysis based on renewable availability time series can only be carried out for the two extreme interconnection settings energy islands and copperplate (compare Section 2.1).”

(Remarks on code availability)

Reviewer #3

(Remarks to the Author)

I enjoyed reading this work, which is highly policy relevant and contributes to the understanding of long-duration storage needs in future energy systems. The analysis is well-structured and provides valuable insights into the role of interconnection, nuclear power, and storage in mitigating renewable energy droughts.

However, certain key methodological limitations, while acknowledged by the authors, could be explored in greater depth to strengthen the scope and impact of this work.

Below, I provide some suggestions to strengthen the manuscript further.

I would encourage the authors to present the impact of nuclear power on long-duration storage needs as a main result, rather than treating it as a robustness check. Additionally, I suggest clarifying the assumptions regarding nuclear's ability to operate as a backup for variable renewable energy (VRE) during long-term droughts. Given nuclear's inflexible nature, it would be helpful to explicitly discuss whether this assumption aligns with real-world operational constraints. Finally, the statement that “moderate, policy-oriented levels of nuclear power in the capacity mix of five European countries mitigate long-duration storage needs across Europe only to a minimal extent” could be reworded to emphasize the key finding more clearly. Since this result is potentially relevant for policymakers, refining the language could enhance its impact.

I would advise the authors to analyze wind and solar droughts separately in some robustness runs, to quantify their individual contributions to long-duration storage needs. This would provide valuable insights into which resource is the dominant driver of system vulnerability. I suspect wind is the primary factor, but a clear quantitative assessment would strengthen the analysis.

The authors mention that hydro and pumped storage reduce long-duration storage needs, but I encourage them to discuss whether hydropower droughts were considered in the model. If these were not included, I would recommend adding a discussion on the potential consequences of their exclusion for the validity of the results.

I suggest evaluating how Scenario 2 and 3 compare to current interconnection levels. To enhance the robustness of the analysis, I recommend introducing an additional scenario that represents current interconnection levels as a baseline for comparison.

While I appreciate the focus on a zero-emission power mix by 2050, I believe it would be valuable to explore how the system would behave if a limited amount of fossil-based generation were available for flexibility. I suggest illustrating the trade-off between relying on low-emission but costly storage solutions versus cheaper, more flexible fossil backup, while also quantifying the environmental impact.

The authors briefly acknowledge the limitations of using historical weather data, but I encourage a more substantial discussion of this issue. There is an emerging body of research on climate-driven changes in wind and solar generation, including datasets such as those published here:

<https://www.nature.com/articles/s41597-024-04129-8>

<https://www.nature.com/articles/s41597-023-02494-4>

I recommend citing these works and elaborating on the potential limitations of disregarding climate change in long-term energy planning. If feasible, I would strongly encourage incorporating future VRE projections into the analysis to enhance the study's policy relevance.

The authors mention among the limitations the lack of detailed demand modeling. I suggest expanding on this by discussing the uncertainties related to summer cooling demand, particularly in Southern European countries such as Spain. There is growing empirical evidence suggesting that cooling demand may increase significantly, which could influence storage and flexibility requirements, see eg:

<https://www.nature.com/articles/s41598-023-31469-z>

The discussion of Figure 4 is insightful, but I recommend extending it by illustrating how optimal long-duration storage varies by country. This would help readers identify which regions benefit most or least from increased interconnection.

The authors note that considering imperfect foresight could offer additional insights. I encourage them to explore whether

their modeling framework allows for any sensitivity analysis on this aspect. If this is not feasible within the current study, I recommend including a discussion of how imperfect foresight could alter the results.

I find Section 3.4 and parts of the results discussion difficult to follow due to insufficient explanation of technical details. I would advise the authors to clarify these sections, particularly for a broader audience that may not have deep technical expertise.

Overall, this paper is an important contribution to the discussion on long-duration storage and energy system resilience. I believe the above refinements would further strengthen its clarity, relevance, and policy impact.

(Remarks on code availability)

No remark

Version 1:

Reviewer comments:

Reviewer #1

(Remarks to the Author)

The authors have sufficiently addressed my previous comments, and the quality of the manuscript has improved considerably. The paper now meets the standards of Nature Communications. I therefore recommend it for acceptance.

(Remarks on code availability)

Reviewer #2

(Remarks to the Author)

1.

I seem to be in a minority in thinking that this study, while technically good, provides a very incremental contribution to the literature.

For example, Shaner et al (E&ES, 2018) wrote in their abstract:

“However, to reliably meet 100% of total annual electricity demand, seasonal cycles and unpredictable weather events require several weeks’ worth of energy storage and/or the installation of much more capacity of solar and wind power than is routinely necessary to meet peak demand.”

<https://pubs.rsc.org/en/content/articlelanding/2018/ee/c7ee03029k>

2.

If the authors do choose to report numbers to three significant digits, they definitely need to qualify those sentences with a clause or adjective indicating that this is in the real world. Nobody could have that degree of precision in talking about the real world.

Aside from concerns related to incrementalism, my main outstanding concern relates to the way model results continue to be presented as if they directly apply to the real world.

Typically, in a scientific paper, the Results section describes the quantitative Results from the model, and the Discussion section discusses the relevance of those model results to the real world.

The authors of this paper do not respect this distinction very well, and discuss the model results as if they were talking about the real world, despite the myriad of factors and uncertainties that are present in the real world but not their model.

I would advise starting Section 2 with a sentence saying, “In this section we discuss our model results. In section 3 (Results), we discuss the relevance of our model results to the real world.

For example, in the Abstract, lines 17-19, they write:

“Assuming policy-relevant interconnection, we find 351 TWh long-duration storage capacity or 7% of yearly electricity demand optimal to deal with the most extreme event in Europe.”

First, the idea that the could project hours of long-duration storage to three significant digits is on the face of it absurd.

Second, determination of real-world optimal approaches also depends on considerations such as political feasibility, various

environmental considerations, jobs, etc. Suggest “least-cost” as an alternative.

Suggest something like:

““In our model, assuming policy-relevant interconnection, we find ~350 TWh long-duration storage capacity, ~7% of yearly electricity demand, in the least-cost system that can deal with the most extreme event in Europe.”

Another example is found on lines 118-120:

“Figure 1 illustrates drought patterns of renewable technology portfolios with policy-relevant capacity mixes from the Ten Year Network Development Plan (TYNDP) 2022, which are retrieved from the renewable time series analysis.”

The naïve reader might think they are look at looking at actual results. Suggest:

“Figure 1 illustrates drought patterns of simulated renewable technology portfolios with policy-relevant capacity mixes from the Ten Year Network Development Plan (TYNDP) 2022, which are retrieved from the renewable time series analysis.”

Similarly, the caption for Figure 1:

“Figure 1: Drought events, electricity demand, and state-of-charge of long-duration storage in winter 1996/97.”

Should read (as I understand it the drought events and electricity demand are based on real data):

“Figure 1: Drought events, electricity demand, and simulated state-of-charge of long-duration storage in winter 1996/97.”

The caption for Figure 4 is:

“Figure 4: Optimal long-duration storage energy capacity aggregated across all countries for all modeled weather years and interconnection scenarios.”

I suggest:

“Figure 4: Long-duration storage energy capacity aggregated across all countries in least-cost optimizations for all modeled weather years and interconnection scenarios.”

An example of this failure to separate discussion of the model with discussion of the real world can be found on lines 232-237: (Note also “This results” should be “These results” or “This result”).

“This results suggests that 159 TWh (equivalent to 3.2% of the yearly European electricity demand) are the minimum need for long-duration storage capacity in a fully renewable European energy system. However, since unconstrained geographical balancing cannot be realized in reality, results of scenario (3) suggest that policy-relevant long-duration storage capacities for coping with the most extreme Dunkelflaute in the data are much higher, amounting to 351 TWh or 7.1% of the yearly demand.”

This should be:

“In our model, 159 TWh (equivalent to 3.2% of the yearly European electricity demand) are the minimum needed for long-duration storage capacity in a fully renewable, reliable, European energy system. However, since unconstrained geographical balancing cannot be realized in reality, scenario (3) may be our most policy-relevant results for long-duration storage capacities needed to cope with the most extreme Dunkelflaute in the data. Long-duration storage capacity in this scenario are much higher, amounting to 351 TWh or 7.1% of the yearly demand.”

There are other places deeper in the paper where this same kind of change needs to be made.

The point is that, given all of the uncertainty in the world, it is absurd to say that a simple model like this can predict optimal amounts to three significant digits. You run a model and get quantitative results, and those results might inform what is good to do in the real world, but the results do not suggest we need to build 159 TWh or 351 TWh. I do not know what real-world uncertainty is. We are not sure what dispatchable power will be available, how much load shifting can be expanded, etc. We do not know future technology costs. We have not done any analysis of near optimal systems so we do not know the cost of having 200 TWh, 300 TWh or 400 TWh of long-duration storage available.

We may want to tell decision makers that we will likely need several hundred TWh of long duration storage to make a reliable system, but it does not add to credibility to say that we have determined we will need 351 TWh of storage. That is just an absurd understanding of decision-making under uncertainty with learning.

Please understand that your model results and what might be good to do in the real world are likely two different things.

(Remarks on code availability)

Version 2:

Reviewer comments:

Reviewer #2

(Remarks to the Author)

I apologize for my inexcusable delay in re-reviewing this manuscript.

The paper seems to be technically fine.

However, my main concern is novelty.

I acknowledge that nobody has done exactly what this paper does, but the idea that we need several weeks' worth of energy storage to produce reliable low-cost VRE-based electricity system has been well established.

Further, several papers have used costs in electricity models to identify resource droughts. Specifically, reference [51]

Grochowicz, A., van Greevenbroek, K., & Bloomfield, H. C. (2024). Using power system modelling outputs to identify weather-induced extreme events in highly renewable systems. *Environmental Research Letters*, 19(5), 054038. <https://doi.org/10.1088/1748-9326/ad374a>

I still have minor issues with some tone and do not find "This is common practice, others do it" to be a valid justification.

The pretense to accuracy from a simple model run under conditions of deep uncertainty and many simplifying assumptions is disconcerting to me. That others exhibit similar pretense does not seem a good justification. One might have made assumptions regarding future cost uncertainties, demand uncertainty, etc, and done Monte Carlo simulations, and come up, I am sure, with very wide uncertainty ranges on estimates.

That said, the question of novelty, is a question of taste, about which well-informed reasonable people can differ.

(Remarks on code availability)

Point-by-point response for “Coping with the Dunkelflaute: Power system implications of variable renewable energy droughts in Europe”

Manuscript reference: NCOMMS-25-00631

Reply to reviewer #1:

This is a novel and valuable piece of work on the management of renewable energy droughts, especially considering the increasing need for large-scale deployment of weather-dependent renewables in decarbonized power systems as a core element of climate mitigation strategies.	We thank the reviewer for taking the time to review our paper and for providing many insightful comments. We have aimed to address all of them in the revised version of the manuscript. We are confident that this has further enhanced the quality of the article. Please find detailed point-by-point responses in the table below.
Unlike many previous studies, this research offers three key contributions • Application of a New Drought Mass Metric: This study applies a new drought mass metric by integrating the Variable-Duration Mean-Below-Threshold (VDMT) method with multiple drought thresholds. Although the approach is based on the authors’ previous work, it is more comprehensive than most other studies on the same topic, which typically rely on a single arbitrary threshold for drought identification.• Combining Drought Identification with Energy System Modeling: Unlike most prior work that analyzes droughts solely from the supply side of VRE generation and in isolation, this paper combines drought identification methods with energy system modeling. This integration allows for an in-depth analysis of long-term (hydrogen-based) storage requirements and their operation under weather year variability within an energy system context.• Assessment of Different Grid Configurations: The study considers various grid configurations, including autarky, copperplate, and limited cross-border capacity. This enables an examination of the effects of virtual storage, which can help mitigate the impacts of renewable energy droughts.	We are very happy that the reviewer shares our view of the potential relevance of our work for the research community and thank her or him for the positive assessment.
However, the paper also presents a few shortcomings or arguable methodological choices, particularly related to the selection of assumptions, which should be addressed or further clarified by the authors.	We thank the reviewer for raising a range of constructive comments, which we address in detail in the following.
The authors use seasonal-agnostic thresholds (i.e., based on fractions of the long-term mean capacity factors) to identify droughts. However, this approach does not account for seasonality and, therefore, cannot distinguish between low production events caused by regular seasonal dips and those driven by true weather anomalies	Thank you for this challenging remark. We agree that it is of interest to differentiate between regular, seasonal variations and extreme events of low renewable availability. Seasonal thresholds could enable this differentiation by correcting for seasonal variations. Two examples of recent analyses with time-varying thresholds, which we also cite in our manuscript, are the following:

or extremes. Employing seasonal thresholds is arguably crucial to improve the efficiency of identifying potentially worst weather years that contain the most extreme droughts, which are critical for power system operations.

- Stoop et al. (2024), <https://doi.org/10.1088/1748-9326/ad27b9>
- Antonini et al (2024), <https://doi.org/10.1038/s43247-024-01260-7>

However, in our opinion, such seasonal thresholds pose methodological challenges on the one hand, and their usefulness for identifying extreme events that have the largest power system impacts remains questionable on the other.

As for methodological challenges, implementing seasonal thresholds is not straightforward, as seasons differ across and within countries, between weather variables (e.g., temperature, solar irradiation, or wind speeds). The approaches used in the above-mentioned articles detect renewable droughts as deviations from a reference profile that reflects seasonality, such as the climatological mean hourly profile over many years. For each hour of the year, this metric indicates an hour's mean value across those many years. Pronounced cumulative deviations, often defined over user-specified time horizons, are then referred to as extreme drought events, which can be characterized by either long duration, very pronounced deviations, or both.

Concerning the usefulness of seasonal thresholds, it remains in our opinion unclear how deviations from a climatological mean profile of a renewable technology portfolio relate to energy system needs. A moderate cumulative deviation occurring in winter and coinciding with peak demand periods could exert much higher system stress and require substantially more system flexibility than a pronounced cumulative deviation occurring for instance in fall, spring, or summer. What matters, especially for long-duration storage needs, is the cumulative energy deficit, i.e., electricity demand net of VRE supply. These energy deficits can be very pronounced during extreme droughts. Further, the choice of the time horizon over which cumulative deviations are detected remains user-specific (cp. Kittel & Schill, 2024, <https://doi.org/10.1088/2753-3751/ad6dfc>).

The cumulative nature of our multi-threshold drought mass metric allows for identifying extreme drought events characterized by long-duration, extremely low renewable availability, or both. This is because the metric considers drought events identified by a wide spectrum of thresholds that reflect increasing levels of drought severity. It thus emulates the concept of finding pronounced energy deficits while abstracting from concrete demand profiles, which very much depends on the specific configuration of the energy system. To account for demand seasonality, which is generally more pronounced in winter in Europe and a relevant driver for long-duration storage needs (compare Section 2.1, 2.2, SI.3), we compute not only the drought mass for the entire summer-to-summer planning horizon of the capacity expansion model, but also for winter droughts occurring between October and

	March, Note that we also consider and illustrate extreme summer droughts, e.g., in Figure 1 and 5. In summary, we are thus confident that our multi-threshold approach is suitable for identifying extreme renewable drought periods that are relevant for long-duration storage needs without requiring seasonally differentiated thresholds.
Ultimately, the primary motivation for drought identification is to enhance the efficiency of weather year selection for energy system modeling and, consequently, to support the planning of a more resilient energy system—especially when dealing with large samples of weather year data.	Thank you for this valuable comment. We fully agree with the reviewer’s perspective. Highly detailed models used to explore decarbonization pathways in large interconnected systems, such as Europe, often face computational limits. This restricts or even prevents modeling of many weather years, which is essential to identify weather-resilient configurations for systems with high shares of variable renewable energy. For example, the decarbonization scenarios for 2050 in the Ten-Year Network Development Plan by the European Transmission Operators rely on just a single weather year. Our paper demonstrates that a pre-selection of critical years with severe renewable droughts, such as 1996/97, is feasible through an analysis of renewable availability time series.
The installed capacity allocation of VRE assets/technologies within and across countries is used as a weighting factor to derive country-wide generation profiles (as in the autarky case) and EU-level generation profiles (as in the copperplate case). These generation profiles are subsequently used for drought identification, capacity expansion, and impact assessment. However, the capacity allocation is primarily based on a single scenario—the TYNDP 2022 'Distributed Energy' scenario—which may affect the robustness of the quantified results. It is recommended to incorporate capacity allocations from alternative sources to assess the sensitivity of the results.	We thank the reviewer for pointing out the potential benefits of including alternative assumptions on the wind and solar PV capacity layout. To address this point, we conducted an additional sensitivity analysis (see supplementary information SI.3, which uses the capacity assumptions of the 'Global Ambition' scenario of the TYNDP. The wind and solar capacity assumptions of this scenario vary substantially: The 'Distributed Energy' scenario (our baseline parameterization) features more solar PV, while the 'Global Ambition' scenario includes a higher share of offshore wind. We illustrate the results in the new Figure SI.5. It shows that the alternative capacity assumptions only have a minor impact on long-duration storage outcomes. Overall, the regression graphs show that the findings are qualitatively similar. In some countries, and also in the copperplate scenario, the drought mass of the largest yearly events tends to decrease. This is because of less solar PV in the capacity mix, which alleviates combined wind and solar drought events in wintertime. However, the slopes of the regression lines hardly change. We further find that the year 1996/97 still leads to the highest drought mass for the copperplate scenario.
The near-term demand profiles are adopted but scaled to the 2050 demand levels based on the TYNDP 'Distributed Energy' scenario. However, the 2050 demand levels also account for the electrification of the space heating and cooling sectors, which would inevitably result in a different demand profile compared to the near-term one. This approach may introduce consistency issues, and the authors should clarify the underlying rationale.	We thank the reviewer for pointing out the need to clarify the rationale for using these particular demand profiles. The main reason is that we aimed to rely on a consistent data source for all time series used in the analysis, including availability factors of onshore and offshore wind power and solar PV, as well as demand. The demand profile is then scaled to the projected 2050 demand level of the mentioned TYNDP scenario. A more detailed consideration of the demand profiles of heating, cooling and other sector coupling technologies as well as their potential flexibility interactions with long-duration storage appears to be beyond the scope of our analysis. Considering that near-term demand profiles include less electricity for heating (heat pumps), electric vehicles and green

hydrogen production than assumed in the 2050 TYNDP scenarios, the linear scaling is likely to cause some distortions. While there is no standard procedure for demand scaling in the literature, we can assume that future electric vehicles and electrolyzers will at least partly be operated in a system-oriented way, i.e., a disproportionate share of their electricity consumption will be during hours with low residual load. While this has implications especially for peak (residual) load, it appears less relevant for long-duration storage needs. We further assume that the future electricity consumption of electric vehicles and hydrogen production is unlikely to have a pronounced seasonal pattern.

Electric heating, however, has a very seasonal load pattern in Europe. Here, our scaling approach likely underestimates the future electricity consumption of heat pumps especially in winter, where the most extreme renewable energy droughts are also located. Accordingly, we are likely to underestimate long-duration electricity storage needs in our analysis.

We would like to point out that our demand profiles already do contain some electric heating, varying by country. Accordingly, they already show some seasonality (compare Figure 3 and section 4.4). For example, in France the amount of electric heating included in the near-term demand time series is already substantial.

What is more, we also do not model heat storage, which might mitigate long-duration electricity storage needs to a substantial extent. Similarly, winter heat demand peaks could be mitigated by energy-efficient building renovations (Zeyen et al. 2021). We expect that these factors will balance each other to some degree with respect to potential distortions of long-duration electricity storage outcomes.

While a brief discussion of these potential distortions of model outcomes was already included in the initial submission, we have extended the respective discussion section 3 of the revised version.

Reply to reviewer #2:

This paper examines the need for long-duration energy storage in the context of a wind and solar reliant European electricity system.	We thank the reviewer for taking the time to review our paper and for providing many insightful comments. We have aimed to address all of them in the revised version of the manuscript. We are confident that this has further enhanced the quality of the article. Please find detailed point-by-point responses in the table below.
While I could quibble about some of the choices made, overall, the analysis is performed at a high level of quality. The paper is well written and organized. The results are consistent with a large body of work.	We are very happy about this positive overall assessment.
The main question I have about this study is the degree to which it is novel. The main concern about how they express their findings is the degree to which they appear to make claims about the real world based on a limited number of model simulations under a specific set of assumptions.	We thank the reviewer for these challenging remarks. We have carefully considered your comments and analyzed the provided literature. Please refer to the boxes below, where we thoroughly address these individually. We have further revised our entire manuscript to make sure that our findings are presented in the context of our analysis' setting to avoid inappropriate generalization.
This paper concludes that several weeks worth of long-duration storage are needed to cost-effectively provide a reliable electricity system that is reliant on wind and solar generation. This conclusion is consistent with a large number of other studies reaching similar conclusions (e.g., Albertus et al., 2020; https://www.cell.com/joule/fulltext/S2542-4351(19)30539-2).	We thank the reviewer for mentioning this useful reference, which we have included in the revised version. We would like to note that Albertus et al. use a specific definition of the term “long-duration storage”, i.e. storage durations between 10 and 100 hours, and refer to technologies with larger storage durations as “seasonal storage”. We do not follow this terminological suggestion in our paper but use the general term “long-duration storage” also for durations beyond 100 hours. Further, the focus of the cash flow analysis presented by Albertus et al. (2020) is on the fact that long-duration storage requires low energy capital costs (\$/kWh) to be viable and does not directly relate to the fact that several weeks of storage are required for energy systems based on variable renewables. To address the reviewer’s remark on “a large number of other studies reaching similar conclusions”, we have carefully checked the references mentioned in Albertus et al. (2020) and found that these often relate to older studies where the shares of renewable energy sources are sometimes substantially lower than assumed in our analysis, or to shorter-duration storage, and/or to smaller energy systems than modeled here. Compare, for example, the article by Denholm and Mai (2019) mentioned therein. Nonetheless, we have revisited the literature and added a few more references in the revised version, which we deem relevant and which should help to give a better account of the current state of knowledge and the contribution we aim to make with our work. This includes: - Ziegler et al. (2019), who investigate storage needs and storage energy cost targets for selected locations and different types of load profiles in the US.- Lux et al. (2022), who determine hydrogen storage needs for Germany.

	 - Chu et al. (2024), who also highlight the role of long-duration storage and its interaction with short-duration storage for a case study of California. - Hunter et al. (2021), who compare lifetime costs for different storage technologies and corroborate our statement that geologic hydrogen storage currently appears as a promising storage option. We have added these references in the literature section of the revised version. We would like to note that these previous analyses are limited either in terms of the geographic or temporal scope or the renewable share as compared to our analysis. Furthermore, we go substantially beyond previous analyses by systematically investigating how the largest renewable energy droughts in Europe (measured via the drought mass), correlate with long-duration storage needs (determined by a capacity expansion model).
They also conclude that there is a tradeoff between long-duration storage requirements and the geographic integration of electricity grids. This conclusion also has been reached by a number of previous studies (e.g., Staadecker et al., 2024; https://www.nature.com/articles/s41467-024-53274-6).	We thank the reviewer for this remark and the additional reference. We agree that previous studies have shown that there is a trade-off between long-duration storage requirements and the geographic integration of electricity grids. We would like to highlight that we never made the claim of novelty with respect to that specific point in our paper. We are nonetheless confident that our analysis can still add substantial and important insights to the relationship between storage and grid for the European context. For instance, Staadecker et al. (2024) focuses only on the West of the US and only considers a single weather year. Similarly, Brown et al. (2021) focuses only on the US, and considers seven different weather years. With respect to Europe, Roth and Schill (2023) consider twelve countries with ten weather years, while Ihlemann et al. (2022) draw on two weather years and five countries. In contrast to that, our analysis includes all of Europe and covers a multitude of weather years. What is even more important, we focus specifically on the impact of pan-European renewable drought events on storage needs, which none of the previously mentioned papers do explicitly. We thus believe that our analysis generates additional insights into how storage needs are formed in an interconnected energy system through extreme renewable drought events, based on a large weather data set, and therefore advances the knowledge on the relationship between temporal and geographic balancing. To better reflect our contribution, we have added the following sentence in the conclusion: “While previous literature has found a similar effect in case studies covering parts of the US or Europe and a limited selection of weather years (Staadecker et al., 2024, Brown et al. 2021, Roth and Schill 2023, Ihlemann et al. 2022}, we show that this finding also holds for a pan-European renewable energy system of 33 countries, considering the most extreme drought events occurring in a large data set of 35 weather years.” Staadecker, M., Szinai, J., Sánchez-Pérez, P.A. et al. The value of long-duration energy storage under various grid conditions in a zero-emissions future. Nat Commun 15, 9501 (2024). https://doi.org/10.1038/s41467-024-53274-6

	Patrick R. Brown, Audun Botterud, The Value of Inter-Regional Coordination and Transmission in Decarbonizing the US Electricity System, Joule, Volume 5, Issue 1 (2021). https://doi.org/10.1016/j.joule.2020.11.013 Ihlemann, M., Bruninx, K., & Delarue, E., Exploring the trade-off between long-term storage deployment and transmission expansion in the power sector. In 2022 18th International Conference on the European Energy Market (EEM) (pp. 1-5). IEEE (2022). https://doi.org/10.1109/EEM54602.2022.9921025 Roth, A., Schill, W.-P., Geographical balancing of wind power decreases storage needs in a 100% renewable European power sector, iScience, Volume 26, Issue 7 (2023). https://doi.org/10.1016/j.isci.2023.107074.
They also note that long-duration storage technologies can also meet short-duration storage needs. This conclusion too has been reached in other studies (e.g., Li et al., 2024; https://pubs.acs.org/doi/full/10.1021/acs.est.3c10188)	We thank the reviewer for this thoughtful and challenging remark, as well as for pointing out the relevant literature. Li et al. (2024) analyze the cost-reducing effects of various storage technologies that differ in (dis-)charging and energy capacity costs, as well as storage duration. Their study models greenfield scenarios for stylized fully renewable energy systems in the US, which are characterized by wind and demand seasonality and abstracts from expansion constraints for wind and solar power. They find that long-duration storage enables the integration of larger amounts of wind power surplus over extended timescales at lower costs compared to shorter-duration storage, which would otherwise require curtailment. This leads to a significant reduction in the wind power capacity required to meet demand, resulting in the largest system cost reductions. Long-duration storage becomes increasingly competitive as wind shares rise. Moreover, they show that long-duration storage can be used to meet short-duration discharging requirements, thus adding more value to systems with high seasonality, if those storage technologies are mutually exclusive. However, in scenarios where both long- and short-duration storage options are admissible, a cost-optimal system configuration deploys both technologies. Our analysis complements and goes beyond the one presented by Li et al. (2024). First, we model the European energy system, which features pronounced wind, solar, and demand seasonality, along with a highly interconnected transmission grid and hydrogen demand from coupled sectors. Second, we incorporate policy-relevant expansion limits for wind and solar energy in accordance with the 2024 Ten-Year Network Development Plan (TYNDP) conducted by the European transmission system operators. Third, we model a partial brownfield scenario, where capacities for run-of-river, reservoirs, open- and closed-loop pumped hydro, and biomass are fixed, while wind, solar PV, and battery storage are optimized endogenously. In this setting, significant deployment of solar PV necessitates both short- and long-duration storage in the cost-optimal system. Our study further extends the analysis in Li et al. (2024) by investigating the operational patterns and interactions of not only storage options but a broader set of shorter- and long-duration flexibility options. These include battery storage, biomass, hydro reservoirs, and pumped hydro for short-duration flexibility. First, we show that the combined use of

	these options reduces the need for electrolysis capacity to charge long-duration storage before extended renewable droughts, as well as the discharging capacity required during those events, thus enabling a least-cost system design. Second, we demonstrate that long-duration storage can also reduce the need for short-duration discharge capacity by helping balance diurnal solar PV variability, particularly in regions like Spain, which experience lower solar seasonality and have access to mid-duration storage. Third, we identify a “merit order” of storage technologies during pronounced renewable droughts. We fully acknowledge the relevance of Li et al. (2024) in this context and have added the following sentence to the revised section 2.4: “These findings complement those of a previous study focused on the US context, which demonstrated that long-duration storage can also provide short-term flexibility (Li et al., 2024).” A similar reasoning applies when comparing our study with Chu et al. (2024), which we mention above in a previous comment.
This paper seems to be professionally done and adds incrementally to our understanding. The main thing that is a little disconcerting about this paper is the degree to which it makes very strong and specific claims based on a very limited number of stylized simulations, while appearing to claim generality.	We thank the reviewer again for the overall positive assessment of the paper’s quality and for providing many helpful comments. We would like to highlight that this paper substantially advances the knowledge of renewable energy droughts and their impact on a decarbonized European power system. To our knowledge, this nexus has not yet been studied in that depth. Therefore, we believe we can make a contribution to the existing literature that is more than only incremental. Like every model-based analysis, its results are only true in the context of the assumptions made. We are aware that our model necessarily includes some stylized elements, but the overall scenario design builds on a policy-relevant scenario framework of the TYNDP. In addition, our model has a large geographic (30 countries) and temporal (35 weather years) which goes beyond most of the existing literature. While our results are conditional on a range of model and parameter assumptions, we believe they offer relevant conclusions for European system design debates. Nonetheless, we acknowledge the concern regarding the generality of the findings. We agree that it is useful to qualify our claims in the sense that concrete, quantitative insights of course apply to our specific model setting. In consequence, we went through the entire paper to edit our statements accordingly. In many instances, we added expressions such as “in our setting” to make it explicit for the reader that we talk about model results, and not real-world phenomena. However, we also want to highlight that the model and its results lie at the heart of this paper. Therefore, we believe that we do not need to qualify every single statement, as most of them relate to model results. However, we made sure that conjectures from the model to the real world were made more explicit.
Specifically, costs of various technologies are likely to vary in the future, with many of them decreasing (e.g., Hunter et al., 2021; https://www.cell.com/joule/fulltext/S2542-4351(21)00306-8). The	Thank you for this valuable comment. We fully agree that the ratio of wind and solar capex to long-duration storage energy capacity capex is decisive for the optimal deployment of these technologies, particularly in the context of long-lasting renewable energy droughts. We therefore included a sensitivity analysis that varies the capex of wind and solar capacity as well as long-duration storage energy in the revised version

specific amounts of long-duration storage in their optimizations will be sensitive to cost assumptions, yet there are no sensitivity studies looking at how results might be sensitive to cost. Long-duration storage is used only because it is cheaper than building more capacity, so the amount of long-duration storage “needed” is a function of the costs of the various technologies and not a generalizable fact.	of the manuscript. Since practical experience with underground hydrogen storage facilities is limited, we vary the costs for long-duration storage across a wider spectrum than for wind and solar capacity. Results show that higher storage energy capex as well as lower wind and solar capex lead to lower optimal capacities of long-duration storage energy, while renewables are overbuilt. However, this substitution effect is limited due to the assumed policy-relevant capacity bounds (in line with the TYNDP2024). A more detailed description and a respective graph are provided in Section 2.6 of the revised manuscript. To further address the reviewer’s comment, we have also revised many instances in the paper where we mentioned that storage capacities would be “needed” or “required” and rephrased this to “optimal” capacities or storage investments. For example, in the introduction we have changed the statement “We further investigate how varying levels of geographical balancing via interconnection affect long-duration storage needs” to “We further investigate how varying levels of geographical balancing via interconnection affect optimal long-duration storage investments”.
In this light, the authors make a very quick jump from their model simulations to the real world, and talk as if their model results apply directly to the real world. Whenever they give numerical results they should be clear they are talking about their model, and not make the assumption that their model results directly apply to the real world. The authors are aware of the simplifications and assumptions inherent in their analysis, and thus should be cautious about stating specific quantitative results as they apply to the real world. An example of this tonal issue can be seen in the abstract, where they write: “Assuming policy-relevant interconnection, long-duration storage of 351 TWh or 7% of yearly electricity demand is required to deal with this event.” What they mean to say is: “Assuming policy-relevant interconnection, in the model configuration used here (including our specific cost assumptions, etc), the least-cost solution deploys 351 TWh (or 7% of yearly electricity demand) of long-duration storage to deal with this event.”	As mentioned in the reply to the comment above, we have rephrased notions of capacity “needs” or “requirements” in many instances. In doing so, we have also mentioned the specific context and parameterization of our quantitative analysis several times. For example, at the end of the introduction we now write: “Here we combine two open-source methods to investigate how VRE droughts impact optimal long-duration storage investments in a fully decarbonized European power sector.” We have also revised the abstract accordingly. The respective passage now reads: “A long-duration storage capacity of 351 TWh or 7% of yearly electricity demand is optimal to deal with this event in our parameterization, assuming policy-relevant interconnection.” However, we would like to highlight that the findings of quantitative energy system analyses are generally only valid in the particular setting analyzed. This is not specific to our study. Typically, real-world conclusions are still drawn from context-specific analyses in the literature, which in our opinion is both justified and useful, under the condition that the modeling framework and the assumptions made suggest policy-relevance. In our case, we are confident that our scenario assumptions, which are based on a policy relevant scenario of European network operators, a large weather data set, and a broad geographic scope of overall Europe, combined with our open-source model analysis using a well-established tool, sufficiently justify drawing relevant real-world conclusions for Europe. In our opinion, adding too many repetitive disclaimers throughout the paper would thus neither be necessary, nor would it improve the flow of reading.
I do not see any discussion of value of lost load or unmet demand. What are	In supplementary information SI.6, we included an additional sensitivity analysis that investigates the impact of a wide range of value of lost

the maximum dual values across all time steps and nodes? This would indicate the shadow cost of electricity in a perfectly efficient market. In many optimizations, this value can be extraordinarily high. Do the model results vary substantially if a “reasonable” value of lost load is used?	loads. Results show that load shedding may have limited effects on optimal capacity deployment of wind and solar, batteries, and long-duration reconversion. These effects are substantially less pronounced for higher value of lost load levels, which appear to be more realistic (Kachirayil et al., 2025). Notably, neither total system costs nor long-duration storage energy capacity, which is at the heart of our analysis, change significantly across all investigated values of lost load. Given these results and the extent of additional analysis in the revised version of the manuscript, we did not include an analysis of dual values of the energy balance.
They say “In a sensitivity analysis, we add a generic firm zero-emission generation technology.” Very clearly, if this technology were very low cost, no long-duration storage would be needed. If it were very costly, then introduction of this generation technology would have no effect on solutions. I don’t know how hard it is to run the model, but it would be interesting to sweep through costs of this generic dispatchable generator and see how long-duration storage needs vary with the cost of this generator.	We thank the reviewer for this remark. We fully agree that there is a trade-off between variable renewables combined with long-duration storage and firm zero-emission generation technologies which depends on their relative costs. The cheaper the latter becomes, the higher its share in the least-cost solution. However, please note that we did not endogenize the capacity of nuclear power in our sensitivity but rather based it on the “Distributed Energy” scenario of the TYNDP. In the revised version of the manuscript, we have added another sensitivity analysis with a substantially higher nuclear capacity based on the “Global Ambition” scenario of the TYNDP (please also see reply to remarks made by reviewer #3). Results indicate that optimal long-duration storage investments decrease further compared to the previous sensitivity with lower nuclear capacity. Still, a substantial amount of long-duration remains optimal. Please also note the additional sensitivity for Germany in the supplemental information, where we sweep through a large range of capacity assumptions and show that long-duration storage remains part of the optimal capacity mix as long as VRE technologies are still present, even for very high capacities of firm zero-emission generation. We would further like to refer to a recent study carried out by the Germany Academies of Science, which investigated the role of firm low-carbon baseload plants (Stöcker et al. 2025). This study finds that low-carbon baseload technologies only become competitive against variable renewables and long-duration storage under very optimistic cost assumptions way below current trajectories. It also finds that overall model outcomes, including long-duration storage, do not change much. Large-scale firm generation capacities rather interact with hydrogen imports to Europe, which are increasingly substituted with domestic production when low-carbon baseload generation gets cheaper. We have also referenced this source in the discussion section of the revised version.
Further, often the potential surface that defines the optimal solution is very broad, and technology mixes that are far from the optimal mix might increase system cost only modestly. Given the large number of uncertainties regarding cost, permitting, demand, etc., it is useful to think about near-optimal solutions (c.f. Neumann and Brown, 2021; https://www.sciencedirect.com/science/article/pii/S0378779620304934).	We agree with the reviewer that using near-optimal solutions within energy system modeling is a very useful technique to understand the solution space of optimization methods. It would probably be a good idea to apply such methods as often as possible. In this paper, we refrain from exploring near-optimal solutions because of three reasons. First, the number of viable alternatives to long-duration storage is very limited. Therefore, the solution space might not be as flat as in other applications. We also would like to highlight that we are in principle agnostic to which technology will be serving the energy storage needs and do not want to favor a single technology, but rather focus on the energy and power capacities needed to serve the load at all times.

	Second, our analysis already includes numerous scenarios, in which we vary the degree of interconnection, the amount of nuclear power and also the availability of other dispatchable technologies that could substitute long-duration storage. Although we do not explicitly try to find near-optimal solutions, we implicitly shed light on the solution space and the robustness of our findings. This is probably most relevant for the nuclear (or more generally, zero-emission firm generation) sensitivity, as it is the only conceivable competitor for long-duration storage in our setting. Please note that we have substantially extended the nuclear sensitivity compared to the initially submitted manuscript, as mentioned in answers to comments above and below. Third, performing a full-fledge near-optimal solution analysis would also go beyond the scope of our paper, which is already rather rich. Given the points made above, we thus believe that this analysis does not lend itself to including a near-optimality investigation. However, we do not want to exclude the usefulness of such analysis for future research, potentially in a follow-up paper.
Minor points: It would be useful to have a schematic diagram showing what energy technologies are considered in the model. The schematic wiring diagram could represent the entire electricity grid as a single node, and the technologies considered by different boxes. This would let the interested reader quickly see which technologies are being considered.	We thank the reviewer for the good remark. Based on this suggestion, we have expanded the existing figure SI.1: Schematic overview of the hydrogen module in DIETER such that it depicts all generation and storage technologies, as well as demand and cross-country flows for both electricity and hydrogen. The figure has been moved to section 4 “Methods” to provide an overview of the model.
This sentence needs some expansion (page 6): “Note that VRE drought analysis based on renewable availability time series can only be carried out for the two extreme interconnection settings energy islands and copperplate (compare Section 2.1).”	Thank you for this indication. We adapted the manuscript as follows: “By design, the VRE drought analysis based on renewable availability time series does not account for limited cross-border exchange. Consequently, it is conducted only for the two extreme interconnection scenarios: (1) energy islands and (4) an idealized pan-European copperplate. In the latter case, the renewable availability time series of all countries are aggregated into a weighted single pan-European composite time series (compare Section 4.1).”

Reply to reviewer #3:

I enjoyed reading this work, which is highly policy relevant and contributes to the understanding of long-duration storage needs in future energy systems. The analysis is well-structured and provides valuable insights into the role of interconnection, nuclear power, and storage in mitigating renewable energy droughts.	We thank the reviewer for taking the time to review our paper and for providing many insightful comments. We have aimed to address all of them in the revised version of the manuscript. We are confident that this has further enhanced the quality of the article. Please find detailed point-by-point responses in the table below.
However, certain key methodological limitations, while acknowledged by the authors, could be explored in greater depth to strengthen the scope and impact of this work.	Thank you for this suggestion. We agree that improvements in this regard can enhance the paper and have added respective material in the revised version. Please see comments below for further details.
I would encourage the authors to present the impact of nuclear power on long-duration storage needs as a main result, rather than treating it as a robustness check.	We thank the reviewer for highlighting the importance of nuclear power for long-duration storage outcomes. Following the reviewer’s suggestion, we have moved the description of the results of the scenario runs with nuclear power to the main results section. The new subsection “2.5 Impact of firm zero-emission generation” deals in detail with that topic. We took the opportunity to revise the entire section and have added another scenario with higher nuclear capacities. In the updated manuscript, the scenario “low levels of nuclear power” assumes 24 GW (as before), while “high levels of nuclear power” assumes 102 GW across Europe. The new Figure 7 shows the hourly and daily generation patterns of nuclear power aggregated across all countries in the weather 1996/97 in both additional scenarios. The figure indicates extended periods of nuclear generation at full capacity in winter to deal with extreme renewable droughts. The revised Figure 8 highlights the impact of different levels of nuclear power capacities on aggregate long-duration energy storage capacities, depending on the level of interconnection. Low levels of nuclear power reduce optimal long-duration storage energy capacity across all investigated interconnection scenarios only to a minor extent. In contrast, higher nuclear capacities reduce the reliance on long-duration storage to a larger extent, especially in case of limited interconnection.
Additionally, I suggest clarifying the assumptions regarding nuclear’s ability to operate as a backup for variable renewable energy (VRE) during long-term droughts. Given nuclear’s inflexible nature, it would be helpful to explicitly discuss whether this assumption aligns with real-world operational constraints.	We thank the reviewer for this comment. In fact, the question of how flexibly nuclear power plants can generate electricity, and if their operational constraints limit their ability to serve as backup for variable renewables, has been much discussed in the German energy policy space. We include minor costs for ramping down or up related to wear and tear and potentially also to energy losses. Other than that, we abstract from inter-hourly operational constraints and basically assume that nuclear power plants are sufficiently flexible to complement the hourly variable renewable fluctuations that occur in our model parameterization. The new Figure 7 illustrates the hourly (vertical axis) and daily (horizontal axis) operation of nuclear power plants in the model. We would like to note that this rather optimistic assumption on the operational flexibility of future nuclear power plants is not only backed by real-world experiences, such as the load-following operation of nuclear plants in France, but also in line with large parts of the literature, compare

	for example the papers by Sepulveda et al. (2018) or Stöcker et al. (2025), both cited in the revised manuscript. Accordingly, our findings on nuclear’s potential to mitigate long-duration storage investments can be interpreted as an upper bound. Strikingly, even under our optimistic assumptions on nuclear flexibility, its effect on long-duration storage remains limited, as illustrated in section 2.5. In reality, we expect that the major limiting factor for the future deployment of nuclear power plants is not a lack of operational flexibility, but rather their high investment costs and respective risks for potential investors. We have mentioned our optimistic flexibility assumptions in section 2.5: “We assume that nuclear power serves as a firm zero-emission generation capacity that does not face inter-hourly operational constraints or but minor costs for ramping up or down that relate to wear and tear costs and potentially also to energy losses.” We have also added a brief discussion of their potential impacts in section 3.1 of the revised version of the manuscript that encapsulates the previous reasoning: “As we largely abstract from inter-hourly flexibility constraints, nuclear's potential to substitute long-duration storage needs as found here can be interpreted as an upper bound of this storage-mitigating effect. Including more specific operational constraints would likely reduce it. Notably, even under our optimistic assumptions on the flexibility of nuclear power, its effect on long-duration storage remains very limited.”
Finally, the statement that “moderate, policy-oriented levels of nuclear power in the capacity mix of five European countries mitigate long-duration storage needs across Europe only to a minimal extent” could be reworded to emphasize the key finding more clearly. Since this result is potentially relevant for policymakers, refining the language could enhance its impact.	Thanks for the comment. Based on our additional scenario runs with different capacities of nuclear power (see comments above), we have revised this statement and have additionally discussed of the role of nuclear energy in section 3.
I would advise the authors to analyze wind and solar droughts separately in some robustness runs, to quantify their individual contributions to long-duration storage needs. This would provide valuable insights into which resource is the dominant driver of system vulnerability. I suspect wind is the primary factor, but a clear quantitative assessment would strengthen the analysis.	We think that separately investigating wind and solar droughts is an excellent idea. However, we feel that this would be way beyond the scope of the present analysis. In fact, we have carried out extensive and technology-specific drought analyses for wind and solar power in various European countries in a companion paper (Kittel and Schill 2024, arXiv 2410.00244, also referenced in the manuscript). There the analysis is mainly based on renewable energy time series and the drought mass metric. Carrying out a respective investigation in the research framework of the present manuscript, i.e., using the capacity expansion model, would not only add substantial length to paper, which is already rather rich; it would also raise conceptual questions of how to attribute long-duration storage needs to specific renewable technologies. One way forward could be to start with the optimal generation capacities of wind or solar power as determined here, and then exogenously increase or decrease these

	capacities and investigate the effects on long-duration storage. We think this could be a promising approach for a potential follow-up paper. While we think providing “a clear quantitative assessment” is out of scope, we still aim to qualitatively address the reviewer comment in the revised manuscript. On the one hand, we have added an illustration of drought patterns of solar PV, onshore wind power and offshore wind power for three selected countries and the European copperplate in the SI (Figure SI.2). This Figure indicates that the largest droughts in renewable energy technology portfolios (panel a) are in fact the result of coinciding drought events of individual technologies (panels b-d) in wintertime. These droughts then drive storage needs. On the other hand, a new sensitivity introduced in section 2.6 sheds light on the contribution of wind and solar power to the most extreme drought events. We repeat the time series-based analysis with an alternative TYNDP scenario (“Global Ambition”) that features a lower share of solar PV but a higher share of offshore wind power as compared to our baseline renewable technology portfolio. Under these assumptions, we find that the drought mass of the most extreme events tends to decrease in some countries and also in the copperplate scenario (See also Figure SI.5). This suggests that wind power alone does not drive the largest drought events and respective storage investments, but that coinciding solar energy droughts play a substantial role.
The authors mention that hydro and pumped storage reduce long-duration storage needs, but I encourage them to discuss whether hydropower droughts were considered in the model. If these were not included, I would recommend adding a discussion on the potential consequences of their exclusion for the validity of the results.	We thank the reviewer for raising this important point. In fact, we do not include hydropower droughts in our analysis. Hydropower inflow time series are neither part of the drought identification tool, nor are they varied between years. Hence, we do not claim to capture any impact of hydropower droughts on results. To justify this simplification, we would like to point out that many countries do not possess significant hydropower capacities and only in a few countries, the capacity mix is strongly influenced by hydropower (e.g., Norway, Switzerland, Austria). While analyzing hydropower droughts would indeed be very interesting, we feel that including a full-scale analysis would be beyond the scope of this paper, which appears to be already rather long. Considering the limited overall contribution of hydro power to the overall European electricity supply in our parameterization we would further argue that including hydro droughts is unlikely to strongly alter our results. Inspired by the reviewer’s comment, we aim to extend our drought identification tool and include run-off-river droughts and the inflow to other hydropower technologies in future work. Following the recommendation of the reviewer, we added the following sentence to the discussion section of the paper: [...] “Finally, we do not account for hydropower droughts, neither in the time series analysis nor in the capacity expansion model. Analyzing hydropower droughts and their interactions with wind and solar droughts appears to be a promising avenue for future research. At the same time, it appears unlikely that including hydro droughts would qualitatively alter our main findings, as the number of affected countries and the overall contribution of hydro power to European electricity supply remains limited.” [...]

I suggest evaluating how Scenario 2 and 3 compare to current interconnection levels. To enhance the robustness of the analysis, I recommend introducing an additional scenario that represents current interconnection levels as a baseline for comparison.	We thank the reviewer for sharing this idea. While we agree that adding another scenario with current interconnection levels could generate additional insights, we do not think that it would constitute a better baseline for comparison. Please note that we largely adopt a greenfield perspective in our model analysis anyway and also abstract from several other legacy capacities in the power sector. We further do not expect relevant additional insights concerning our research question from this scenario. We expect that storage capacity in the optimal solution of a setting with current interconnection levels would increase compared to (2), while remaining lower than (1). This is because of the limited potential of geographical balancing of pronounced droughts. Considering the limited potential for additional insights, the number of already implemented interconnection scenarios, the increased length of the paper after the revisions, and the fact that current interconnection levels will anyway not be present in the long-term scenarios modeled here, we decided to refrain from including this additional interconnection scenario. However, following the recommendation of the reviewer, we added the following sentence to the results section of the paper: [...] “If the expansion of the electricity or hydrogen networks in scenarios (2) or (3), as envisaged by European transmission system operators, cannot be realized, our results suggest that optimal long-duration storage energy capacities will be between those of scenarios (1) and (2).” [...]
While I appreciate the focus on a zero-emission power mix by 2050, I believe it would be valuable to explore how the system would behave if a limited amount of fossil-based generation were available for flexibility. I suggest illustrating the trade-off between relying on low-emission but costly storage solutions versus cheaper, more flexible fossil backup, while also quantifying the environmental impact.	We thank the review for this critical remark. We agree that a fossil backup technology is a relevant option and should be investigated in the context of this analysis. We therefore conducted an additional sensitivity analysis, we introduces oil-fired backup capacity combined with direct air carbon capture and storage for carbon removal. Results show that only in the case of extremely low direct air carbon capture of 100EUR/t CO₂, this backup technology can substantially reduce long-duration storage energy capacity, while total cost decrease about four per cent. Given current knowledge, it appear unlikely that such costs can be realized (Young et al., 2023). For scenarios assuming higher costs, these effects remain insignificant. Given the uncertainty of future direct air carbon capture and storage, we conclude that the role of fossil backup capacity for dealing with extreme renewable droughts remains limited in a climate-neutral European energy system.
The authors briefly acknowledge the limitations of using historical weather data, but I encourage a more substantial discussion of this issue. There is an emerging body of research on climate-driven changes in wind and solar generation, including datasets such as those published here: https://www.nature.com/articles/s41597-024-04129-8 https://www.nature.com/articles/s41597-023-02494-4	The reviewer highlights a very interesting aspect: the impact of climate change on future availabilities of solar and wind. There is indeed already a substantial and growing number of papers that have created datasets and analysis on that topic. We fully share the view that it would be exciting to incorporate future VRE availability projections in renewable energy drought analyses. In fact, we already plan to do so in a follow-up paper. In the present manuscript, we feel that incorporating additional renewable time series that reflect future climate change, and investigating potential differences to historic data, would go way beyond the scope of this paper. This would also include making an appropriate selection of time series from a multitude of climate model projections, e.g., from CMIP6. This would be relevant, given that climate models often differ in design, resulting in potentially varying models outcomes for similar scenarios.

I recommend citing these works and elaborating on the potential limitations of disregarding climate change in long-term energy planning. If feasible, I would strongly encourage incorporating future VRE projections into the analysis to enhance the study’s policy relevance.	However, given the variety of climate models, this is beyond the scope of this paper. Instead, this certainly merits a dedicated analysis in a stand-alone paper – or several ones. We have slightly expanded the last paragraph of the paper, in which we highlight this avenue for future research, and have also mentioned the two dataset references suggested by the reviewer there. Additionally, there is already a substantial body of literature that highlights how climate change will generation of renewable, extreme periods, and optimal system configuration in Europe. With respect to overall renewable energy generation, Jerez et al. (2015, https://doi.org/10.1038/ncomms10014) suggest that solar PV generation might slightly decrease in Europe, especially in Northern countries, while slight increases in Southern countries are found. Overall, generation profiles will remain stable enough not to fundamentally change PV generation in Europe. Hou et al. (2021, https://doi.org/10.5194/esd-12-1099-2021) comes to similar conclusions. van der Most (2025, https://doi.org/10.1088/2752-5295/add613) estimates wind and PV variability does not increase strongly due to climate change, yet different regions in Europe will be affected in different ways. With respect to the length of energy droughts periods, they also show varied outcomes. While the Baltic, British Isles, and Eastern Europe experience no change in median drought length, they find that Central and Northwestern Europe will see a slight increase, while Northern Europe experiences a reduction in median drought length. Similar, but slightly different results are found by Kapica et al. (2024, https://doi.org/10.1016/j.rser.2023.114011) who also estimate an increases in wind drought duration in in Central Europe, the UK, and Ireland. Solar energy droughts are expected to increase in parts of Western, Central, and Eastern Europe, while expected to decrease in in Southern Spain, Italy, Greece, and Turkey. They point out that a mix of solar and wind energy (“hybridization”) can help limiting the impacts of extreme weather events. Yu and Vautard (2025, https://doi.org/10.1088/1748-9326/adb2a9) come also the conclusion that the likelihood of wind energy droughts has been increasing lately recent years and will be further increasing due to climate change. The effects are likely to differ between different parts of Europe. Zheng et al. (2024, https://doi.org/10.1038/s41467-024-48966-y), relying on past data, show that climate change has been increasing the duration of extreme low-reliability events worldwide.
The authors mention among the limitations the lack of detailed demand modeling. I suggest expanding on this by discussing the uncertainties related to summer cooling demand, particularly in Southern European countries such as Spain. There is growing empirical evidence suggesting that cooling demand may increase significantly, which could influence storage and	We thank the reviewer for this comment. We think this is a relevant point, but cooling demand is likely to have repercussions on short-duration flexibility technologies rather than on long-duration storage. We have expanded the discussion section with a brief statement which also includes the suggested reference in the revised version of the manuscript: “Similarly, electricity demand is likely to increase in summer in many European countries because of a growing demand for cooling (Colelli et al. 2023). While this may increase flexibility requirements of the future energy system, we expect that it would have a larger effect on short-duration flexibility options than on long-duration storage, considering typical diurnal

flexibility requirements, see eg: https://www.nature.com/articles/s41598-023-31469-z	cooling patterns and their partial coincidence with solar PV generation peaks. Our analysis also indicates that the most pronounced renewable energy droughts, which also drive long-duration storage needs, occur in winter and not in summer in an interconnected European energy system.”
The discussion of Figure 4 is insightful, but I recommend extending it by illustrating how optimal long-duration storage varies by country. This would help readers identify which regions benefit most or least from increased interconnection.	We thank the reviewer for the good remark. Following the suggestion, we have added Figure SI.12 that shows the storage energy capacities in all countries for different scenarios. We find that the decreasing effect of interconnection on long-duration energy storage can be found in most countries. In addition to the figure and text in the SI, we have added the following sentence to the results section of the paper: [...] “Higher levels of interconnection capacities reduce the need for long-duration storage energy capacity both on an aggregate level and in most countries (Figure SI.12)” [...]
The authors note that considering imperfect foresight could offer additional insights. I encourage them to explore whether their modeling framework allows for any sensitivity analysis on this aspect. If this is not feasible within the current study, I recommend including a discussion of how imperfect foresight could alter the results.	The reviewer touches on a very relevant point. The model used in this analysis is deterministic and hence operates under perfect foresight. It is unfortunately not possible to implement scenario runs with imperfect foresight within our current research design, as this would require fundamental changes to the model setup. However, as storage optimization under imperfect foresight becomes more prevalent in recent literature, also in in our research group, we see good prospects for dedicated analysis in the future. We have accordingly revised the respective outlook statement in the second-last paragraph of the paper as follows: “And while perfect foresight within a year results in relevant benchmark outcomes, considering imperfect foresight for long-duration storage operations within a year may offer complementary insights into real-world storage use.” Following reviewer’s suggestion, we have further added the following paragraph to the limitations section of the paper: “[...] Further, the model used in this analysis is deterministic and assumes perfect foresight. Under limited foresight, long-duration storage might be operated more precariously compared to our analysis, i.e., stockpiling more energy over longer time spans and avoiding very low states of charges to hedge against extreme drought events that might occur later in planning horizon (Schmidt, 2025). Accordingly, the optimal operational and capacity decisions of the long-duration storage found in this analysis should be interpreted as benchmark solutions, which may not be achievable by real world actors with imperfect foresight. [...]”
I find Section 3.4 and parts of the results discussion difficult to follow due to insufficient explanation of technical details. I would advise the authors to clarify these sections, particularly for a broader audience that may not have deep technical expertise.	We thank the reviewer for the important remark. Based on this suggestion, we have substantially edited and streamlined section 3.4 (now 2.4), so it reads easier and less technically. Specifically, we have combined several examples, improved our explanations of how flexibility options differ in terms of techno-economic characteristics. In addition, we have edited the entire results section for readability and technical terms, which, we believe, left to a more accessible text.

Overall, this paper is an important contribution to the discussion on long-duration storage and energy system resilience. I believe the above refinements would further strengthen its clarity, relevance, and policy impact.	We thank the reviewer again for the positive assessment and agree that the paper's quality benefited substantially from implementing many of the valuable suggestions.
--	---

Point-by-point response for “Coping with the Dunkelflaute: Power system implications of variable renewable energy droughts in Europe”

Manuscript reference: NCOMMS-25-00631A

Reply to reviewer #1:

The authors have sufficiently addressed my previous comments, and the quality of the manuscript has improved considerably. The paper now meets the standards of Nature Communications. I therefore recommend it for acceptance.

We thank the reviewer for taking the time to review our revised paper and for the positive recommendation.

Reply to reviewer #2:

1. I seem to be in a minority in thinking that this study, while technically good, provides a very incremental contribution to the literature.

For example, Shaner et al (E&ES, 2018) wrote in their abstract:

“However, to reliably meet 100% of total annual electricity demand, seasonal cycles and unpredictable weather events require several weeks’ worth of energy storage and/or the installation of much more capacity of solar and wind power than is routinely necessary to meet peak demand.”

<https://pubs.rsc.org/en/content/articlelanding/2018/ee/c7ee03029k>

We thank the reviewer for taking the time to review the revised version of our manuscript, and for providing additional and insightful comments.

Regarding our contribution, we would like to defend our claim that our analysis substantially adds to the previous literature. Please note the reviewers #1 and #3 strongly support our view.

In particular, we analyze how the most extreme renewable energy droughts impact the need for long-duration storage in Europe in different interconnection scenarios. We are not aware of previous work that similarly investigates the interplay of spatial and temporal system flexibility for coping with extreme European drought events. We further contribute by linking renewable drought analyses that are purely based on renewable availability time series with energy system analysis that determines least-cost storage needs, using capacity expansion models. Here too, we are not aware of previous studies that leverage these insights of such tools. On top of this, we explore the interaction of long-duration storage with shorter-duration flexibility options, nuclear power, or fossil backup plants with emission abatement.

Concerning the mentioned paper by Shaner et al. (2018), their analysis makes a very general point and focuses on the United States. We would like to highlight that we do not claim to be the first to find that long-duration storage is an important component of 100% renewable energy systems. This general finding can in fact be considered common wisdom by now. But our dedicated analysis of least-cost storage capacities for coping with renewable energy droughts in Europe, considering different interconnection states, substantially adds to previously published work, and certainly goes way beyond the analysis presented for the U.S. by Shaner et al. (2018).

Yet, we acknowledge the relevance of the analysis by Shaner et al. (2018) and added it to the literature review section.

2. If the authors do choose to report numbers to three significant digits, they definitely need to qualify those sentences with a clause or adjective indicating that this is in the real world. Nobody could have that degree of precision in talking about the real world.

We thank the reviewer for this important comment. Regarding the reporting of precise results, we agree they should not be used to simulate precisions where there is not any. However, we still argue that it is valid to report precise numbers in this paper, instead of artificially blurring them, as these are valid within the context of our model framework. This seems to be also common practice in most other peer-reviewed papers that report model-based results. We absolutely acknowledge the importance of making sure that the reader understands that these precise results are only valid within our specific modeling framework. To address this, we have revised our

	manuscript, particularly the discussion section, to make a clear distinction between the model results and their implications for the real world.
Aside from concerns related to incrementalism, my main outstanding concern relates to the way model results continue to be presented as if they directly apply to the real world. Typically, in a scientific paper, the Results section describes the quantitative Results from the model, and the Discussion section discusses the relevance of those model results to the real world. The authors of this paper do not respect this distinction very well, and discuss the model results as if they were talking about the real world, despite the myriad of factors and uncertainties that are present in the real world but not their model.	We thank the reviewer for this critical remark. We agree that model outcomes and their applicability should be clearly distinguished, particularly given the fact that all models represent a simplified version of reality. However, we would still argue that energy system modeling, while abstracting from various factors, generates meaningful insights for the real world to provide guidance for system planners and policy makers. We therefore carefully revised the manuscript to make a clear distinction between model results and our interpretation of real-world implications. By doing so, we intend our model-based analysis to provide insights, not concrete projections or forecasts. In the table below, you find examples of these revisions. We further attach a pdf document that visually highlights these revisions.
I would advise starting Section 2 with a sentence saying, "In this section we discuss our model results. In section 3 (Results), we discuss the relevance of our model results to the real world.	We added the following sentence at the beginning of Section 2 (Results): "Here, we discuss our model results. In Section 3, we discuss the relevance of our model results to the real world."
For example, in the Abstract, lines 17-19, they write: "Assuming policy-relevant interconnection, we find 351 TWh long-duration storage capacity or 7% of yearly electricity demand optimal to deal with the most extreme event in Europe." First, the idea that the could project hours of long-duration storage to three significant digits is on the face of it absurd. Second, determination of real-world optimal approaches also depends on considerations such as political feasibility, various environmental considerations, jobs, etc. Suggest "least-cost" as an alternative. Suggest something like: "In our model, assuming policy-relevant interconnection, we find ~350 TWh long-duration storage capacity, ~7% of yearly electricity demand, in the least-cost system that can deal with the most extreme event in Europe."	Again, we appreciate the reviewer's critical perspective. We agree that the abstract should clearly indicate model results. We revised the manuscript, particularly regarding the use of "optimal" and used "least-cost" instead. For instance, we adjusted the stated sentence in the abstract as follows: "Assuming policy-relevant interconnection in our model, we find 351TWh long-duration storage capacity or 7% of yearly electricity demand in the least-cost system that can cope with the most extreme event in Europe." Additionally, we would like to emphasize that reporting key model results in the abstract is common practice in energy systems research, including in leading journals such as Nature Communications. This practice allows readers to grasp the main insights directly from the abstract without having to read the entire paper. Given the large volume of research in this field, we regard this convention as valuable and have therefore retained these figures in the abstract.
Another example is found on lines 118-120: "Figure 1 illustrates drought patterns of renewable technology portfolios with policy-relevant capacity mixes from the Ten Year Network Development Plan (TYNDP) 2022, which are retrieved from the renewable time series analysis."	We thank the reviewer for this comment and have revised the manuscript accordingly. It reads now: "Figure 1 illustrates drought patterns of simulated renewable technology portfolios with policy-relevant capacity mixes from the TYNDP 2022, which are retrieved from the renewable time series analysis."

The naïve reader might think they are look at looking at actual results. Suggest: “Figure 1 illustrates drought patterns of simulated renewable technology portfolios with policy-relevant capacity mixes from the Ten Year Network Development Plan (TYNDP) 2022, which are retrieved from the renewable time series analysis.”	
Similarly, the caption for Figure 1: “Figure 1: Drought events, electricity demand, and state-of-charge of long-duration storage in winter 1996/97.” Should read (as I understand it the drought events and electricity demand are based on real data): “Figure 1: Drought events, electricity demand, and simulated state-of-charge of long-duration storage in winter 1996/97.”	We thank the reviewer for this comment and have revised the caption accordingly. It now reads: “Simulated drought events, electricity demand, and least-cost state-of-charge of long-duration storage in winter 1996/97.”
The caption for Figure 4 is: “Figure 4: Optimal long-duration storage energy capacity aggregated across all countries for all modeled weather years and interconnection scenarios.” I suggest: “Figure 4: Long-duration storage energy capacity aggregated across all countries in least-cost optimizations for all modeled weather years and interconnection scenarios.”	We thank the reviewer for this comment and have revised the manuscript accordingly. It reads now: “Least-cost long-duration storage energy capacity aggregated across all countries for all modeled weather years and interconnection scenarios.”
An example of this failure to separate discussion of the model with discussion of the real world can be found on lines 232-237: (Note also “This results” should be “These results” or “This result”.) “This results suggests that 159 TWh (equivalent to 3.2% of the yearly European electricity demand) are the minimum need for long-duration storage capacity in a fully renewable European energy system. However, since unconstrained geographical balancing cannot be realized in reality, results of scenario (3) suggest that policy-relevant long-duration storage capacities for coping with the most extreme Dunkelflaute in the data are much higher, amounting to 351 TWh or 7.1% of the yearly demand.” This should be: “In our model, 159 TWh (equivalent to 3.2% of the yearly European electricity demand) are the minimum needed for long-duration storage capacity	We thank the reviewer for the critical remark. We adjusted the paragraph in the revised version accordingly. It now reads: “Hence, in our model, 159TWh or 3.2% of the yearly European electricity demand, is the minimum need for long-duration storage capacity for a reliable, fully renewable European energy system. However, since unconstrained geographical balancing cannot be realized in reality, scenario (3) shows the most policy-relevant long-duration storage need that is required for dealing with the most extreme simulated Dunkelflaute event in the data. In this scenario, long-duration storage capacities are much higher, amounting to 351TWh or 7.1% of the yearly demand.”

in a fully renewable, reliable, European energy system. However, since unconstrained geographical balancing cannot be realized in reality, scenario (3) may be our most policy-relevant results for long-duration storage capacities needed to cope with the most extreme Dunkelflaute in the data. Long-duration storage capacity in this scenario are much higher, amounting to 351 TWh or 7.1% of the yearly demand.”	
There are other places deeper in the paper where this same kind of change needs to be made.	Thank you very much for the remark. We have gone thoroughly through the paper, especially the results and discussion section, and have adjusted our wording so that it should become clear to the reader that we discuss model results and not real-world effects. Please have a look at the attached document that shows the direct changes we made to the manuscript. For instance, the opening of section 3 reads now as follows (bold new): “This paper analyzes how extreme renewable energy droughts impact least-cost investments and operations of long-duration electricity storage in a future renewable European power sector by combining renewable time series analysis and an energy system model.”
The point is that, given all of the uncertainty in the world, it is absurd to say that a simple model like this can predict optimal amounts to three significant digits. You run a model and get quantitative results, and those results might inform what is good to do in the real world, but the results do not suggest we need to build 159 TWh or 351 TWh. I do not know what real-world uncertainty is. We are not sure what dispatchable power will be available, how much load shifting can be expanded, etc. We do not know future technology costs. We have not done any analysis of near optimal systems so we do not know the cost of having 200 TWh, 300 TWh or 400 TWh of long-duration storage available. We may want to tell decision makers that we will likely need several hundred TWh of long duration storage to make a reliable system, but it does not add to credibility to say that we have determined we will need 351 TWh of storage. That is just an absurd understanding of decision-making under uncertainty with learning. Please understand that your model results and what might be good to do in the real world are likely two different things.	We thank the reviewer for this critical and challenging remark. Again, we believe that quantitative analyses add value by providing policy guidance. Yet, we agree that our analysis does not account for near-optimal solutions, in which long-duration storage capacities may significantly differ. To reflect this, we added the following paragraph in the discussion: “Importantly, our analysis focuses exclusively on least-cost solutions and does not account for near-optimal system configurations, which could, for instance, exhibit lower implementation barriers or greater societal acceptability. Future work could employ a modeling-to-generate-alternatives approach to explore the extent to which such near-optimal capacity layouts exist and at what additional system costs [75].” We argue, however, that the potential for alternative solutions is limited due to the structure of our research design. In the case of a pronounced renewable shortage, the model cannot expand transmission grid capacities, which would otherwise enable spatial balancing of the drought event. Apart from long-duration storage, the only remaining option available to the model for addressing energy shortfalls is extensive overbuilding of wind and solar capacities. For policy relevance, capacity expansion limits for these technologies are adopted from the TYNDP 2024, further constraining the model’s degrees of freedom to mitigate long-duration storage needs. Moreover, the cost share of long-duration storage in the critical year 1996/97 is below 10%, whereas renewable generation

accounts for around 50% or more of total system costs. Consequently, minor variations in long-duration storage capacity would likely induce disproportionately larger cost changes through alternative renewable deployment. Therefore, significant shifts in long-duration storage are expected only in scenarios with total system costs well above the least-cost optimum. Overall, we are confident that the order of magnitude of our results remains robust and provides meaningful insights.

Revision III for “Coping with the Dunkelflaute: Power sector implications of variable renewable energy droughts in Europe”

Point-by-point response to the reviewers

Manuscript reference: NCOMMS-25-00631A

Reply to reviewer #2:

I apologize for my inexcusable delay in re-reviewing this manuscript. The paper seems to be technically fine.	We thank the reviewer for taking the time to review our manuscript another time.
However, my main concern is novelty. I acknowledge that nobody has done exactly what this paper does, but the idea that we need several weeks' worth of energy storage to produce reliable low-cost VRE-based electricity system has been well established.	We thank the reviewer for this comment and for raising the import point of novelty again. We would like to reiterate that we do not claim novelty for the general finding that long-duration storage is needed in VRE-based electricity systems. As we have explained in the two previous revisions, we contribute to the literature (1) by connecting energy system modelling with an empirical analysis of renewable availability time series, (2) by quantifying long-duration storage needs for coping with extreme Dunkelflaute events in Europe in terms of optimal operation and capacity expansion, and (3) by investigating how different other technological options influence these storage needs, i.e., interconnection, nuclear power, fossil back-up combined with CSS, and load shedding. We are not aware of a similar analysis, with long-duration storage needs driven by renewable droughts at the heart of the analysis – certainly not in the European context, but probably also for other world regions. Additionally, the studies previously referenced by the reviewer focused on the power system in the US, which differs significantly to the one in Europe, e.g., in terms of size, interconnection, or meteorological conditions.
Further, several papers have used costs in electricity models to identify resource droughts. Specifically, reference [51] Grochowicz, A., van Greevenbroek, K., & Bloomfield, H. C. (2024). Using power system modelling outputs to identify weather-induced extreme events in highly renewable systems. Environmental Research Letters, 19(5), 054038. https://doi.org/10.1088/1755-1315/19/5/054038.	We agree with the reviewer that there is an emerging literature strand analyzing power sector or energy system impacts of extreme events based on shadow prices. Grochowicz et al.'s work can be considered as pioneering in this respect and has already inspired additional research in the field. Yet, we would like to highlight that analyzing shadow prices from energy models is only one option for identifying extreme events. There is significantly more literature on renewable droughts analyzing renewable availability data only, or their mismatch with an exogenous electricity demand profile, often referred to as residual or net load (see the literature review in Kittel & Schill, https://doi.org/10.1088/2753-3751/ad6dfc, particularly Figure 1). Grochowicz et al. compare extreme positive residual load events with periods identified through high shadow price periods and show that those do not perfectly align. However, analyzing positive residual load events comes at the cost of having to make assumptions about future demand profiles and flexibility. Due to the ongoing electrification efforts of end-use sectors, determining meaningful profiles is not straightforward. For example, which temporal profile will the load of future electric vehicle fleets have? Extreme positive residual load events are therefore only applicable in the context of the investigated demand profiles. Other demand profiles will lead to another set of identified events. Kittel & Schill (https://doi.org/10.1088/2753-3751/ad6dfc) discuss this extensively, e.g., in Section 4.3.

Please note that we instead combine i) an empirical analysis of renewable availability data to identify renewable droughts with ii) energy system modeling, where the investigated renewable data is used as an input for the modeling part of the analysis. The deployed drought identification method rules out shortcomings of previous studies on renewable droughts, as discussed in Kittel & Schill (<https://doi.org/10.1088/2753-3751/ad6dfc>). Compared to analyses of residual load, analyzing renewable availability data has the advantage of not requiring assumptions about demand profiles and flexibility. Identified events are hence more generally applicable and not specific to a certain demand scenario.

We then directly link least-cost long-duration storage operation and capacity expansion decisions determined by the energy system model with the identified extreme renewable drought events. This is, to the best of our knowledge, a novelty in the literature.

In the main part of the analysis of Grochowicz et al. (approach ii), they focus on shadow prices to find extreme events as well as analyzing and classifying weather regimes that lead to those events. While they use storage values for their line of reasoning and illustrate the operation of long-duration storage for one representative year as one element in Figure 4 (and another one in Figure S12), long-duration storage needs are not at the center of their analysis. In contrast, our analysis particularly addresses the role of long-duration storage for dealing with renewable droughts. This includes optimal operation and capacity layout as well as interactions with other flexibility options, such as interconnection, short- and medium-duration flexibility, nuclear power, fossil backup combined with CCS, and load shedding in least-cost capacity expansion solutions. In this sense, our analysis has a different focus than Grochowicz et al.

Importantly, shadow prices are very specific to the model setup they result from. This includes the set of assumptions used for model parameterization, the technological, regional, and sectorial scope of the model, other model intricacies such technology representation, etc. Our approach to finding extreme events differs fundamentally. We identify renewable droughts from wind and solar availability time series derived from reanalysis data, which does not depend on any energy system model setup. In this sense, the events identified in our analysis are stable, even if we changed our energy system model setup. In a second step, we use these events to explain energy system model outcomes, focusing on long-duration storage needs.

Next, the duration of identified extreme events differs substantially between Grochowicz et al. and our analysis. They identify system-defining periods as episodes of consecutively high shadow prices, lasting on average one week and at most two weeks, based on a user-defined threshold of cumulative system costs. The authors explicitly acknowledge that restricting system-defining events to durations of at most two weeks constitutes a limitation of their approach. Our analysis directly builds on this observation and demonstrates that critical events may persist for substantially longer periods. Specifically, such events can emerge as sequences of renewable droughts which, when considered in isolation, may each last up to two around weeks. However, these sequential droughts can occur within an extended period of below-average renewable availability that spans well beyond two weeks. During such prolonged sequences, multiple discharge episodes can arise in which long-duration storage operates at capacity, driven by individual extreme droughts. In the spirit of Grochowicz et al., these extreme renewable droughts can be classified as system- or storage-defining, as they trigger investments in long-duration storage capable of sustaining discharge over periods that extend well beyond two weeks.

Additionally, Grochowicz et al. model the power sector only, abstracting from energy demands from other sectors. In contrast, our model includes energy demands from transport, industry, and heat provision, even if in a somewhat stylized way. This leads to substantially differing optimal operational and investment decisions, particularly regarding electrolysis, long-duration storage, and hydrogen reconversion. For example, for the

	extreme event in winter 1996/97, central in our analysis, we find least-cost long-duration storage capacities (cp. Figure 4) considerable higher than those found by Grochowicz et al. (cp. Figure S12 in Grochowicz et al.). We show that these storage needs can be reduced by alternative technologies such as nuclear or fossil-backup. Nonetheless, storage needs found in our analysis appear to be one order of magnitude larger, which emphasized the role of long-duration storage for a sector-coupled energy system. Finally, Grochowicz et al. use cost assumptions associated with the year 2030. Our assumptions relate to 2050. Given substantial cost decreases of renewable and flexibility technologies in the past and projected for the future, assuming costs for 2050 likely impacts the optimal capacity layout of the least-cost system configuration, including storage capacities. Given all of the above, we believe our analysis adds significantly to the literature beyond the insights found by Grochowicz et al.
I still have minor issues with some tone and do not find “This is common practice, others do it” to be a valid justification.	We understand that this remark relates to the comments made in the last round of revisions, where you noted, for example: “If the authors do choose to report numbers to three significant digits, they definitely need to qualify those sentences with a clause or adjective indicating that this is in the real world. Nobody could have that degree of precision in talking about the real world.” We then replied: “Regarding the reporting of precise results, we agree they should not be used to simulate precisions where there is not any. However, we still argue that it is valid to report precise numbers in this paper, instead of artificially blurring them, as these are valid within the context of our model framework. This seems to be also common practice in most other peer-reviewed papers that report model-based results.” We would like to make two points here:  1) In fact, it is common practice in the energy system literature to report exact numbers. This is true for virtually all the many papers we referenced in our article. We also think it has a lot of value to report actual outcomes and not to artificially blur results. 2) Having said that, quantitative outcomes of models are of course only valid in the model context in a strict sense. We are sure that all modelers, and also the vast majority of the readers of this article, are fully aware of this. Therefore, we would not find it useful to make restrictive statements for every quantitative result we discuss. At the same time, we fully agree with the reviewer that the distinction between model results and the real world should not be blurred when it comes to deriving policy conclusions. We have thus critically reviewed our paper again and have rephrased some statements in the abstract and in the introduction and conclusion sections. For example, in the conclusion we now write: “This number indicates that there is a lower bound for storage energy capacity required in a fully renewable European energy system. Its magnitude appears sizable, particularly considering that barely any hydrogen-based long-duration storage capacity is currently installed in Europe.” We hope that this addresses the reviewer’s remaining concerns.
The pretense to accuracy from a simple model run under conditions of	Thank you for sharing your concerns. We think that a good part of these remarks, especially relating to accuracy, should already addressed by our response above – and also by the revisions made in the previous two rounds.

deep uncertainty and many simplifying assumptions is disconcerting to me. That others exhibit similar pretense does not seem a good justification. One might have made assumptions regarding future cost uncertainties, demand uncertainty, etc, and done Monte Carlo simulations, and come up, I am sure, with very wide uncertainty ranges on estimates.	On the notion of deep uncertainty over more or less all model parameters, we fully agree. This is a challenge for all model-based analyses, especially such that investigate long-term future scenarios. Of course, more sensitivities or even Monte Carlo simulations would generally be insightful, potentially varying not only all kinds of cost parameters, but also demand projections etc. However, this would not only come at the expense of broad ranges of results, which would be much harder to interpret, but would also substantially add to the length of the paper. Considering that we already have a rather long manuscript, which in fact features various sensitivity analyses, we feel that adding more sensitivities at this point would be too much of a stretch. Having said that, we think that it would be worthwhile investigating the effects of such uncertainties in future work and have added a respective statement in the conclusion: “Likewise, it would be of interest to investigate how uncertainties over future technology cost or demand developments impact storage outcomes, e.g., using Monte Carlo simulations or other techniques for optimization under uncertainty.”
That said, the question of novelty, is a question of taste, about which well-informed reasonable people can differ.	We thank the reviewer for this reconciliatory remark, and again, for taking the time to discuss and challenge our work. We hope that our responses helped to highlight the novelty and distinct contribution of our work.